# A subset of megakaryocytes regulates development of hematopoietic stem cell precursors

Wenlang Lan (ID)[1,2], Jinping Li (ID)[1,2], Zehua Ye (ID)[1,2], Yumin Liu (ID)[1], Sifan Luo (ID)[1], Xun Lu (ID)[1], Zhan Cao (ID)[1], Yifan Chen[1], Hongtian Chen (ID)[1] & Zhuan Li (ID)[1✉]

## Abstract

Understanding the regulatory mechanisms facilitating hematopoietic stem cell (HSC) specification during embryogenesis is important for the generation of HSCs in vitro. Megakaryocyte emerged from the yolk sac and produce platelets, which are involved in multiple biological processes, such as preventing hemorrhage. However, whether megakaryocytes regulate HSC development in the embryonic aorta-gonad-mesonephros (AGM) region is unclear. Here, we use platelet factor 4 (PF4)-Cre;Rosa-tdTomato⁺ cells to report presence of megakaryocytes in the HSC developmental niche. Further, we use the PF4-Cre;Rosa-DTA (DTA) depletion model to reveal that megakaryocytes control HSC specification in the mouse embryos. Megakaryocyte deficiency blocks the generation and maturation of pre-HSCs and alters HSC activity at the AGM. Furthermore, megakaryocytes promote endothelial-to-hematopoietic transition in a OP9-DL1 coculture system. Single-cell RNA-sequencing identifies megakaryocytes positive for the cell surface marker CD226 as the subpopulation with highest potential in promoting the hemogenic fate of endothelial cells by secreting TNFSF14. In line, TNFSF14 treatment rescues hematopoietic cell function in megakaryocyte-depleted cocultures. Taken together, megakaryocytes promote production and maturation of pre-HSCs, acting as a critical microenvironmental control factor during embryonic hematopoiesis.

**Keywords** Megakaryocytes; Hematopoietic Stem Cell Development; Endothelial to Hematopoietic Transition; Hematopoietic Precursors; TNFSF14

**Subject Categories** Development; Haematology; Stem Cells & Regenerative Medicine

## Introduction

Hematopoietic stem cell (HSC) with the capacity for self-renewal and multi-lineage differentiation is at the top of the hematopoietic lineage hierarchy, contributing all the immature hematopoietic progenitor cells (HPC) and mature blood cells. The application of HSC transplantation is one of the widely used clinic curative treatments for blood diseases. However, the insufficient donor resources of HSC limited their application. HSC regeneration in vitro is hopeful to provide an abundance of HSC for clinical application. The elucidation of regulatory mechanism of HSC production and expansion will improve the HSC reprogramming for clinical curative application.

HSCs emerge from endothelial cells (ECs) with hemogenic potential in the aorta-gonad-mesonephros (AGM) region (Medvinsky and Dzierzak, 1996; Muller et al, 1994), which go through a complex conserved process known as endothelial to hematopoietic transition (EHT) by forming hematopoietic clusters (Boisset et al, 2010; Yokomizo and Dzierzak, 2010), including pre-HSC I and pre-HSC II as well as hematopoietic stem/progenitor cell (HS/PC) (Rybtsov et al, 2011). Although pre-HSC I cells are able to maturate into pre-HSC II cells and functional HSCs, pre-HSCs fail to be detected by direct transplantation, which is the gold standard for functional HSCs. Abundance surface markers are screened for enriching hemogenic endothelial cells (HECs) (Hou et al, 2020; Zeng et al, 2019) and pre-HSCs (Li et al, 2017b; Zhou et al, 2016) via transcriptomic sequencing technology, such as single-cell RNA-Sequencing (scRNA-Seq). Interplays of key transcription factors and signaling pathways, including Runx1, Gata2, Gfi1, Notch, BMP, and Ramp2 pathways, are reported to involve in the regulation of HSC development (Bigas et al, 2013; Chen et al, 2009; de Pater et al, 2013; Dzierzak and Bigas, 2018; Lancrin et al, 2012; Yvernogeau et al, 2020). In addition, yolk sac derived-macrophages (Macs) interact with newly forming hematopoietic clusters to promote HSC production and the (pro)inflammatory factors secreted by neutrophils and macrophages regulate HSC function (Espin-Palazon et al, 2014; Li et al, 2019; Mariani et al, 2019). However, the niche regulating HSC emergence and maturation is not yet known completely.

Megakaryocytes (Mks) are large (50–100 μm) in size and rare (about 0.05%) in the mouse bone marrow of adults, arising from HSC via Mk progenitor (MkP) stage (Psaila and Mead, 2019). Emerging evidence has demonstrated that Mks play vital roles in the pathophysiological process, such as platelet production, inflammation, immunity, bone metastasis as well as HSC quiescence (Bruns et al, 2014; Hoover et al, 2021; Rost et al, 2018; Zhao et al, 2014). In the embryo, MkPs are observed firstly from primitive hematopoiesis in the E7.5 yolk sac. One day later, erythroid-myeloid progenitors (EMP)

[1]Key Laboratory of Functional Proteomics of Guangdong Province, Department of Developmental Biology, School of Basic Medical Sciences, Southern Medical University, Guangzhou, China. [2]These authors contributed equally: Wenlang Lan, Jinping Li, Zehua Ye. ✉E-mail: zhuanli2018@smu.edu.cn

have the ability to generate Mks in the yolk sac, independent of Myb (Tober et al, 2007; Tober et al, 2008; Xu et al, 2001). Noticeably, derived Csf2rb+ fraction of EMPs migrating into fetal liver give rise to Mks, dependent on the Myb (Hoeffel et al, 2015). Previous reports have shown that Mks contain diverse morphologies during developmental stage (Matsumura and Sasaki, 1988). Recently, different groups have reported the heterogeneity of Mks in the embryo and adult bone marrow by means of transcriptomic sequencing technology. The newly identified markers enriched functional or multifaceted Mks, such as a subset of immune response cells (Sun et al, 2021; Wang et al, 2021). Mks are one of the earliest hematopoietic cells, but the physiological roles in HSC development remain yet to be investigated. In this study, we reveal the roles of Mks in HSC development in the mouse embryonic AGM region and imply that one subfraction of Mks promotes hematopoietic cell emergence and mediates the pre-HSC maturation through TNFSF14-Lymphotoxin beta receptor (LTβR) pathways.

# Results

## Platelet factor 4 specifically labels megakaryocytes in the embryonic AGM region

PF4-Cre mouse model is used to check the development of Mks, which is highly expressed in the megakaryocytic lineage (Tiedt et al, 2007). To investigate the specificity, PF4-Cre;Rosa-tdTomato embryos were analyzed by flow cytometric analysis (CD45, CD41, CD42d). In the yolk sac, the percentages of tdTomato+ cells increased from $0.40 \pm 0.04\%$ to $1.08 \pm 0.09\%$ from E9.5 to E11.5. About 90% of tdTomato+ cells were CD42d+ cells in all the detected time points, but only very rare tdTomato+ cells were positive for CD45 (Fig. 1A–C; Appendix Fig. S1A,B). The cell numbers of tdTomato+ cells and tdTomato+CD42d+ cells were increased more than 8 folds between E9.5 and E11.5 (Appendix Fig. S1C). Meanwhile, lower percentages of tdTomato+ cells were observed in the AGM region compared to yolk sac, which also presented the vast majority of Mks. Less than 200 tdTomato+ cells or tdTomato+CD42d+ cells were found in the E9.5-E11.5 AGM regions (Fig. 1D–F; Appendix Fig. S1A,D–G). Exclusively, the tdTomato+ cells were negative in the hematopoietic precursor-related populations including ECs (CD31+CD41-CD45-), pre-HSC I (CD31+CD41lowCD45-), pre-HSC II (CD31+CD45+) as well as CD45+ hematopoietic cells (Fig. 1G,H; Appendix Fig. S1H–J). The immunostaining analysis confirmed that tdTomato+ cells were highly expressed CD41, labeled Mks and platelets in the AGM region, which were enriched more abundant in the dorsal aorta (DA) compared to the area outside of DA (Fig. 1I,J; Appendix Fig. S1K–L), in line with the distribution of Macs (Mariani et al, 2019). However, the percentage of Mks or platelets that interacted with ECs was very low (Appendix Fig. S1M). TdTomato+ signals were positive for Mks in the E10.5 fetal liver by immunostaining (Appendix Fig. S1N) and rare cells were detected positively in the HSC (Lin-Sca1+Mac1lowCD201+) of E12.5 fetal liver and HSC (Lin-Sca1+c-Kit+CD150+CD48-) in E13.5–14.5 fetal liver (Appendix Fig. S1O–S). These data indicate that PF4 mainly labels Mks, but not other hematopoietic lineage cells in this stage of embryos.

To further investigate the roles of PF4 in the development of Mk, PF4-Cre mice were crossed with Rosa-DTA mice to gain PF4-Cre;Rosa-DTA (DTA, deleting cre-expression cells by diphtheria toxin) and control embryos (PF4-Cre or Rosa-DTA, defined as Ctr), identified by genotyping (Appendix Table S1). The live cell numbers were comparable in detected tissues, including DTA yolk sac, AGM region, and fetal liver compared to corresponding control (Fig. 2A; Appendix Fig. S2A,B). In the E10.5 yolk sac, the percentages of CD41+CD45- and CD41+CD42d+ were reduced significantly, but not in the E11.5, however, the number of both fractions was unchanged in the E10.5-E11.5 yolk sac. The reduction in the percentage and cell number of phenotypically defined hematopoietic cells (CD41-CD45+) were observed in the E11.5 yolk sac, indicating the roles of Mks in hematopoietic development (Appendix Fig. S2C–F). Noteworthily, in vitro coculture showed that the ability to produce Mks and platelets was affected in the DTA yolk sac compared to control (Appendix Fig. S2G,H). Furthermore, the percentage and absolute number of CD41+CD45- were reduced by about 30% in the E10.5-E11.5 DTA AGM region, but not in the E9.5 (Fig. 2B,C). Meanwhile, both the percentage and number of CD41+CD42d+ cells were reduced by between 29.6% and 51.2% in the E10.5 and E11.5 (Fig. 2D–G). Similarly, the reduction of CD41+CD42d+ cell number also existed in the E11.5 fetal liver (DTA: $2879 \pm 239$/equivalent embryo (ee) vs Ctr: $3753 \pm 328$/ee) (Fig. 2H,I). PF4 is indeed involved in the Mk development during embryonic stage. Taken together, the deletion of PF4-positive cells is useful for detecting the roles of Mk in hematopoiesis, especially in the AGM region.

## Megakaryocyte deletion reduces hematopoietic stem cell activity, but not hematopoietic progenitor cells

To test whether the Mks are involved in the hematopoiesis, colony forming unit-cultures (CFU-Cs) were performed. In the E12.5 fetal liver, the number of CFU-Cs and CFU-E was not changed (Appendix Fig. S3A). However, the percentage of phenotypically defined HS/PC (Lin-Sca1+Mac1low) and HSC (CD201+Lin-Sca1+Mac1low) were dramatically reduced by about 37% and 58%. More than half of HSCs were diminished in the DTA fetal liver compared with the control group (Fig. 3A–F), indicating that Mks possibly regulate HSC development.

Furthermore, in the earlier stage, the total number of CFU-Cs and CFU-E was not influenced in the E9.5, except for CFU-GM (Appendix Fig. S3B). The obvious reduction of CFU-C number was found in the E11.5 DTA AGM region, with a significant decrease in CFU-GM and the number of CFU-E was reduced in the DTA AGM region. Consistently, a reduced trend was observed in the total CFU-C number of yolk sac (Fig. 3G; Appendix Fig. S3C,D). To further detect the HSC function, AGM cells with 200,000 bone marrow cells were injected into irradiated recipients (Recipients with chimerism ≥5% in peripheral blood are as engrafted). Whereas only 3 out of 6 mice (3/6, chimerism $15.1 \pm 10.0\%$) receiving control AGM cells were engrafted and 1 out of 5 (1/5, chimerism $3.1 \pm 1.0\%$) recipients were engrafted in the DTA group at 4 weeks post-transplantation. After 20 weeks transplantation, no recipients (0/5, chimerism $1.8 \pm 0.9\%$) receiving PF4-Cre;Rosa-DTA AGM cells were with chimerism ≥5%, while 5 out of 6 recipients were engrafted in the control group with the average chimerism $19.0 \pm 12.6\%$ (Fig. 3H), suggesting the possibility of reduced HSC activity. These data support that Mks likely play a critical role in the HS/PC function in the AGM region.

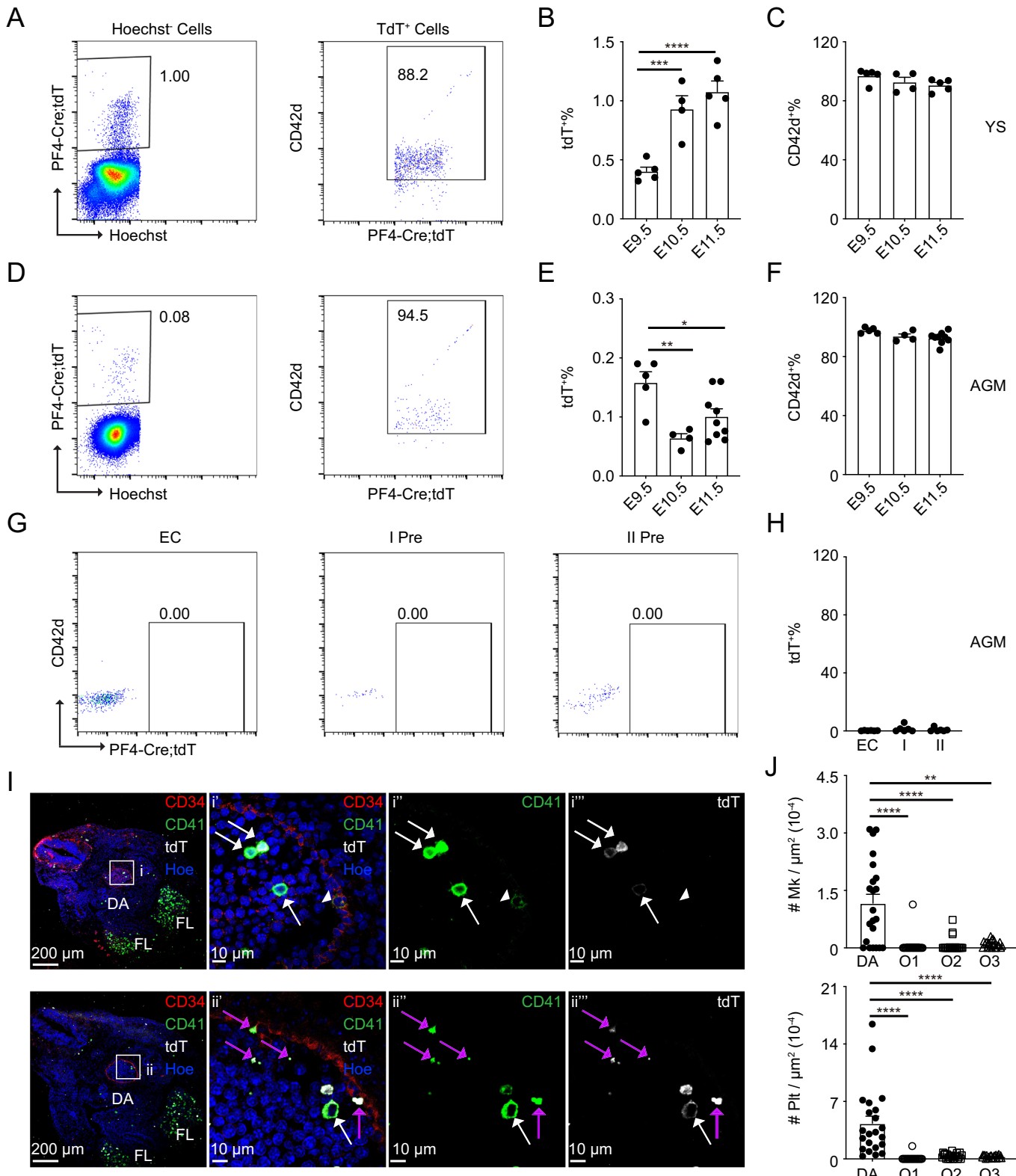

## Megakaryocytes deletion alters pre-hematopoietic stem cell formation and/or maturation

To check whether the Mks are involved in the HSC emergence, flow cytometric analysis was performed to test HEC (CD41⁻CD45⁻ CD31⁺CD44⁺CD201⁺c-Kit⁺), pre-HSC I (CD41$^{low}$CD45⁻CD31⁺CD 201⁺c-Kit⁺) and pre-HSC II (CD45⁺CD31⁺CD201⁺c-Kit⁺). We found that the percentage and cell number of HEC, pre-HSC I, and pre-HSC II were not changed in the E10.5 AGM region after Mk ablation (Appendix Fig. S3E–G). In the E11.5 AGM region, the shrink of Mks

**Figure 1.  Identifying the specificity of PF4 labeling Mks in the embryo.**

(A) Presentative flow cytometric data showing PF4 labels most of Mk in the PF4-Cre;Rosa-tdTomato (PF4-Cre;tdT) yolk sac. (B) The percentages of tdTomato$^+$ Mks in the E9.5-E11.5 Yolk sac. $n = 4$–5 fetuses from at least 3 litters/condition. \*\*\*$p = 0.0009$, \*\*\*\*$p < 0.0001$. (C) The percentages of CD42d$^+$ Mk in tdTomato$^+$ cells in the E9.5-E11.5 Yolk sac. $n = 4$–5 fetuses from at least 3 litters/condition. (D) Presentative flow cytometric data of PF4 labeling most of Mk in the PF4-Cre;Rosa-tdTomato AGM region. (E) The proportions of tdTomato$^+$ Mks in the E9.5-E11.5 AGM. $n = 4$–9 fetuses from at least 3 litters/condition. \*$p = 0.0121$, \*\*$p = 0.0018$. (F) The percentages of CD42d$^+$ Mk in tdTomato$^+$ cells in the E9.5-E11.5 AGM. $n = 4$–9 fetuses from at least 3 litters/condition. (G, H) Flow data displaying PF4 failed to label EC, pre-HSC I, and pre-HSC II in the E11.5 AGM region. $n = 6$ fetuses from at least 3 litters/condition. (I) Immunostaining of cryosections in the E10.5 embryos. Scale bar presents 200 μm or 10 μm. Red = CD34, Blue = Hoechst, Green = CD41, White = tdTomato (tdT). White arrows indicate CD41$^{high}$tdT$^+$ Mks and white arrowheads indicate CD41$^{low}$tdT$^-$ cells. Purple arrows indicate platelets. (J) The number of Mks and platelets per μm$^2$ in the dorsal aorta (DA) and out of DA. O1 = Area (10 μm) out of DA, O2 = Area (50 μm) out of DA, O3 = Area (150 μm) out of DA. $n = 22$ section of 3 fetuses from 3 litters. Statistical significance was determined by Mann–Whitney test. \*\*$p = 0.0014$, \*\*\*\*$p < 0.0001$. tdT = tdTomato, YS = Yolk sac, EC = Endothelial cells, I = pre-HSC I, II = pre-HSC II, FL = fetal liver. Data Information: For all analysis above bars represent mean ± SEM. Statistical significance was determined by unpaired Student's t-test unless the statistical test was indicated. Source data are available online for this figure.

resulted in a dramatic decrease of percentage in pre-HSC I ($0.030 \pm 0.006$‰ vs $0.058 \pm 0.009$‰) and pre-HSC II ($0.140 \pm 0.020$‰ vs $0.220 \pm 0.030$‰), as well as in the HEC ($0.144 \pm 0.018\%$ vs $0.100 \pm 0.014\%$, Fig. 3I–K). Furthermore, the significant reduction of cell number in HECs was examined in the DTA AGM region compared with the control group. Similarly, the cell number of pre-HSC I was reduced by about 50% and about 40% reduction in pre-HSC II (Fig. 3L–N), suggesting that the process of EHT is impaired, especially the formation of pre-HSCs.

As is shown hematopoietic clusters include hematopoietic precursors (Yokomizo and Dzierzak, 2010). Immunostaining data has shown that the total number of hematopoietic clusters (CD34$^+$Runx1$^+$ and/or CD34$^+$c-Kit$^+$) was shrunk obviously after Mk depletion, although the cell number of each hematopoietic cluster was comparable (Fig. 4A,B; Appendix Fig. S3H,I). Then, explant/in vitro cultures and in vivo transplantation were performed to check the function of hematopoietic precursors. In line with results from the fresh E11.5 AGM regions we mentioned before (Fig. 3I–N), the percentage and cell number of pre-HSC I and pre-HSC II was reduced, but not in the HEC fraction of E10.5 AGM after 3 days explant culture (Appendix Fig. S3J,K). OP9 cocultures can study the maturation/differentiation ability of pre-HSCs. Pre-HSC II were sorted and cocultured with OP9 cells for 10 days. Pre-HSC II-derived erythroid cells and Mks were decreased from the DTA group compared to control ones (Appendix Fig. S3L). Meanwhile, the number of hematopoietic cells (CD45$^+$ cells, including myeloid and B lymphoid cells) was much lower in the DTA group than in control, with the reduction of B cells (CD19$^+$ cells) (Appendix Fig. S3M), indicating the impaired maturation/differentiation of pre-HSCs.

Meanwhile, after 4 weeks transplantation, 7 out of recipients (7/8, chimerism $39.0 \pm 11.0\%$) were engrafted in the control AGM explant (AGM$^{ex}$) group, while only 5/9 recipients (5/9, chimerism $27.0 \pm 9.8\%$) were in the DTA AGM$^{ex}$ (1–2 ee/recipient) (Fig. 4C). Around 83% of recipients were reconstituted in the control group, while half of recipients in the DTA group, with much lower chimerism (control: $64.5 \pm 15.0\%$ vs DTA:$25.1 \pm 15.3\%$) and decreased trend in B lymphoid lineage output at 16 weeks post-transplantation (Fig. 4D,E), in line with the changed differentiation ability of pre-HSC II (Appendix Fig. S3M). These data support that Mks are involved in the pre-HSC maturation/differentiation to further affect HSC function.

Abundance data supports that hematopoietic cells including pre-HSCs and HSC emerged from ECs through the EHT process. To further check whether Mks play roles in this process in the

earlier stage, HEC coculture with cytokines ± Mks was performed according to the schematic diagram by using OP9-DL1 system (Fig. 4F). The existence of Mks via 3 days coculture enhanced the hematopoietic cell emergence derived from E10.5 HEC, the number of CD45$^+$ hematopoietic cells produced in the Mix (Mk + HEC) group was significantly higher than that in only HEC group in the E10.5 AGM region, and no CD45$^+$ cells were detected in the Mk group (Fig. 4G). A similar increase phenomenon was observed in the E11.5 AGM (Fig. 4H), indicating that Mks are essential for the process of EHT, which is consistent with the stronger decrease trend in the pre-HSC I compared to that in the pre-HSC II in DTA AGM regions compared to control (Fig. 3J,K,M,N). These data alongside the transplantation data demonstrate that Mks mediate the process of pre-HSC production and their maturation partially in the embryo.

## Transcriptional characterization on the heterogeneity of megakaryocytes

To further characterize the heterogeneity of Mks, single-cell RNA-sequencing (modified STRT-seq) was performed on a total of 384 Mks (CD41$^+$CD42d$^+$CD226$^{+/-}$, 240 cells) from E11.5 AGM regions and EMP cells (CD41$^{low}$c-Kit$^+$, 144 cells) from E9.5 yolk sac (Appendix Fig. S4A), and 351 cells passed rigorous quality control with no batch effect (Appendix Fig. S4B, C). All cells were clustered into 3 clusters, including EMP, Mk1 and Mk2. The bubble graph showed that EMP cluster expressed featured genes Spi1, Ikzf1, and Emb, and genes including Col3a1, Sox11, Lum, and Dcn were expressed in Mk1 and Gp5, Gata1, and Thbs1 in Mk2 cells. We identified 833 genes highly expressed in Mk1 and 1351 genes in Mk2. Noticeably, Mk1 expressed highly Igfbp5, Meg3, Nrp1, and Mk2 expressed Plcg2, Gnaz, Sla2, and Dapp1 (Fig. 5A,B; Appendix Fig. S4D). Metascape analysis showed that enriched genes in Mk1 were related to organ development (heart and head development), neuron development, and negative regulation of cell differentiation, however, higher expressed genes in Mk2 were involved in the regulation of platelet formation/activation, hemostasis, neutrophil degranulation, and cell interactions at the vascular wall (Fig. 5C,D). We further evaluated the cell cycle score of these two clusters and found that a higher percentage of Mk2 cells stayed in the S/G2/M status compared to Mk1, suggesting remarkably activated cycling in the Mk2 (Fig. 5E,F), in line with the gene functional enrichment. Meanwhile, trajectory analysis by Monocle 2 suggested that along the maturation path from EMP to Mk1 and Mk2, Mk2 was in the late stage of developmental dynamics, related to the higher

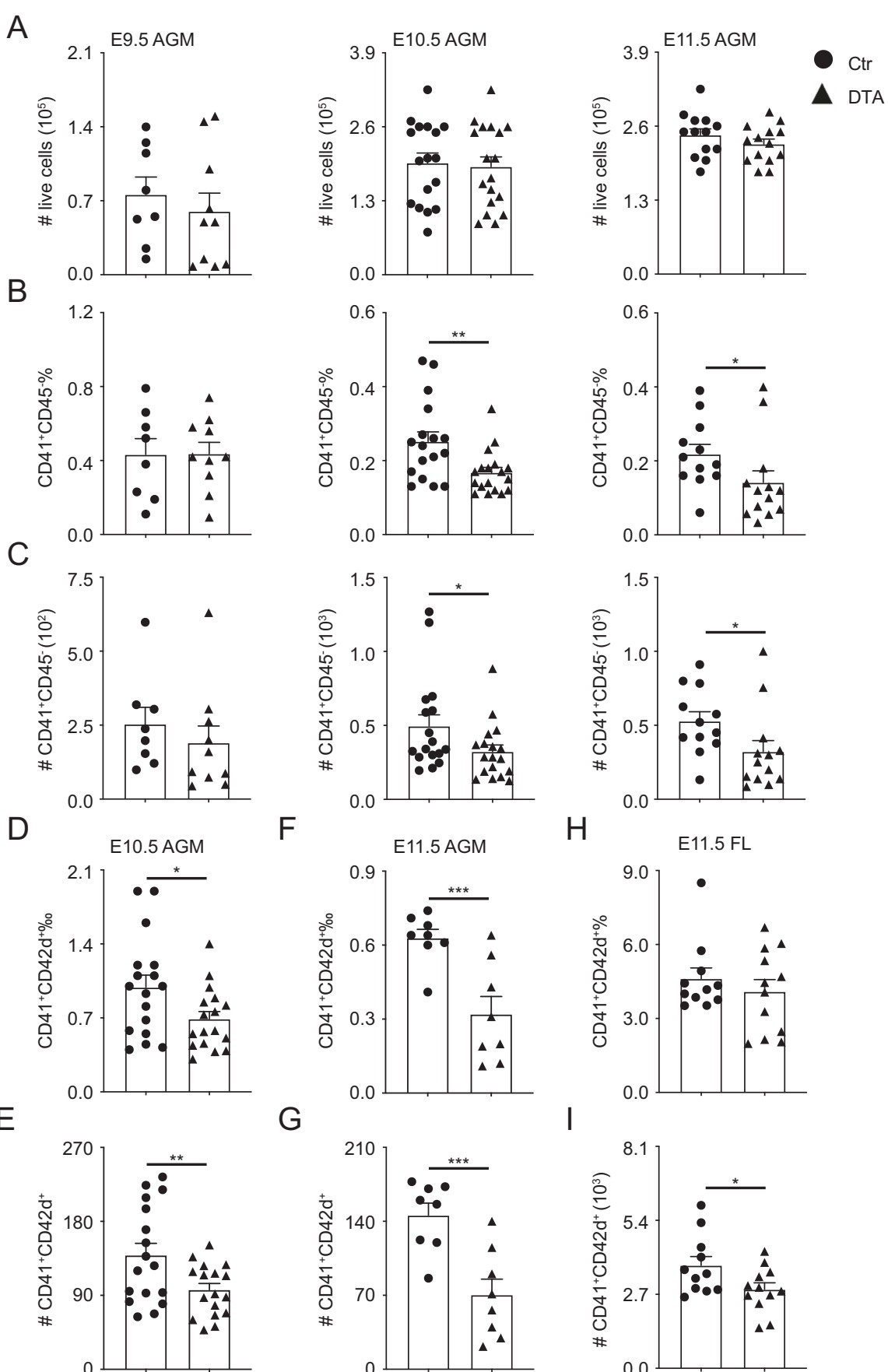

◀ **Figure 2.   Reduction of Mks in the PF4-Cre;Rosa-DTA (DTA) embryos.**

(A) The number of live cells in the E9.5-E11.5 AGM region. $n = 8$–18 fetuses from at least 3 litters/condition. (B, C) Percentages and absolute numbers of CD41$^+$CD45$^-$ cells in the E9.5-E11.5 AGM region. $n = 8$–18 fetuses from at least 3 litters/condition, *$p < 0.05$, **$p = 0.0034$. (D–G) The reduction of CD41$^+$CD42d$^+$ percentages and absolute numbers in the E10.5-E11.5 PF4-Cre;Rosa-DTA AGM region. $n = 8$–17 fetuses from at least 3 litters/condition, *$p = 0.017$, **$p = 0.0071$, ***$p < 0.001$. (H, I) Percentages and absolute numbers of CD41$^+$CD42d$^+$ Mks in the E11.5 fetal liver. $n = 11$–12 fetuses from at least 3 litters/condition, *$p = 0.0204$. Circle = control (Ctr), triangle = DTA. FL = fetal liver. Data Information: For all analysis above bars represent mean ± SEM. Statistical significance was determined by unpaired Student's t-test. Source data are available online for this figure.

expression of *Pf4*, *Gp9*, and *Itga2b* (Fig. 5G,H). The transcription factor (TF) analysis showed the higher expression of TF, for example, *Hhex*, *Tal1*, *Nfe2*, and *Gata1* in the Mk2 (Appendix Fig. S4E, Dataset EV1), which are relative to the Mk maturation or platelet production (Machlus and Italiano, 2013; Pencovich et al, 2011; Shivdasani et al, 1997).

## CD226 is relative to more mature megakaryocytes

In comparison to the gene expression of surface markers between Mk1 and Mk2, 128 surface marker genes were enriched in Mk1, such as *Cdh11*, *Gpc3*, *Col5a1*, and *Fn1*, and around 96 genes were higher expressed in the Mk2, including *Tgfb1*, *Gp5*, *Itgb3*, and *DNAM1* (DNAX accessory molecule-1, *CD226*). CD226 is an immunoglobulin-like glycoprotein, expressed on different hematopoietic cells and some ECs in adults (Reymond et al, 2004; Shibuya et al, 1996; Wagner et al, 2017). The upregulation of *CD226* was also observed by tSNE graph, thus CD226$^+$ and CD226$^-$ clusters were separated according to the expression of CD226 (Fig. 6A,B). Furthermore, the pseudotime analysis showed that CD226$^+$ cluster cells were more mature and most of the CD226$^+$ cluster cells overlapped into Mk2 fraction (Fig. 6C,D; Appendix Fig. S4F). The differences in gene expression and TFs between CD226$^+$ cluster cells and CD226$^-$ cluster cells along with the functional enrichment analysis showed more matured Mks in the former cluster (Appendix Fig. S4G–K; Dataset EV2), demonstrating that CD226 enriches the mature fraction of Mks.

Immunostaining for CD226 showed that more CD226$^+$ Mks were localized mainly inside DA compared to outside of DA (Appendix Fig. S4L–N), similar to the distribution of total Mks (Appendix Fig. S1L). Flow cytometric analysis displayed that $29.8 \pm 2.1\%$ of Mks (CD41$^+$CD42d$^+$) were positive and more than half of Mks were negative for CD226 in the E11.5 AGM region, which is different from E10.5 Mks and similar to E11.5 fetal liver. However, a much higher percentage of CD226$^+$ cells was observed compared to CD226$^-$ cells in the E10.5 yolk sac (Fig. 6E,F and Appendix Fig. S4O–Q). These CD226$^+$ Mk included a higher percentage of big-size cells (with high SSC) than CD226$^-$ Mks in the E10.5–11.5 AGM, E10.5 yolk sac, and E11.5 fetal liver (Fig. 6G; Appendix Fig. S4R–T). Cytospins of E11.5 AGM Mks revealed that CD226$^+$ Mks possessed a ploidy-like nucleus, with rough cell membranes and CD226$^-$ Mks with an oval-round nucleus and smooth cell membranes (Fig. 6H), suggesting CD226$^+$ Mks present late developmental states.

Subsequently, we detected what kind of Mks were affected in the DTA embryos. In the E11.5 AGM region, the percentage of CD226 in the Mks was reduced dramatically in the DTA embryos compared with control group. More than 60% of CD226$^+$ Mks were deleted and the percentage of bigger Mks also decreased by around 60% in the DTA group, a similar trend was observed in the

E10.5 AGM, yolk sac and E11.5 fetal liver (Fig. 6I–K; Appendix Fig. S4U–C1). Importantly, the ratios between CD226$^+$ and CD226$^-$ Mks were decreased in the E10.5-E11.5 AGM region (Fig. 6L). In addition, the reduced ratios were observed in the E10.5-E11.5 yolk sac and E11.5 fetal liver (Appendix Fig. S4D1–E1), demonstrating that the deletion of Mks was mainly on the mature Mks in the DTA embryos. These data suggest that CD226$^+$ Mks are the main fraction to affect hematopoietic development in the AGM region.

## CD226 discriminates functional megakaryocytes in regulating hematopoiesis

To distinguish the function of CD226$^+$ and CD226$^-$ Mks in hematopoietic development, depending on the OP9-DL1 coculture system, HECs (±Mks) cocultures were established and harvested after 3 days for examining the production of CD45$^+$ cells. Cell combinations included: HECs alone, HECs + Mks (CD41$^+$CD42d$^+$, Mix), HECs + Mks$^+$ (CD226$^+$ CD41$^+$CD42d$^+$, Mix$^+$) and HECs + Mks$^-$ (CD226$^-$CD41$^+$CD42d$^+$, Mix$^-$). The existence of CD226$^+$ Mks enhanced higher CD45$^+$ cell production than that in the HEC group ($575 \pm 167$ vs $294 \pm 128$/ee), which is comparable to the Mix group (including CD226$^+$ and CD226$^-$ Mks) ($750 \pm 142$/ee). Nevertheless, CD226$^-$ Mks failed to affect the emergence of CD45$^+$ cells ($286 \pm 71$/ ee) (Fig. 6M). Thus, CD226$^+$ Mks boost the ability of EHT.

Furthermore, to test whether the contact between Mks and HECs is essential for EHT, transwells were used to separate Mks and HECs in the coculture system. We found that the existence of transwell (Mix$^{+T}$: Mix$^+$ + Transwell) was unable to change the enhanced effects of CD226$^+$ Mks on the EHT, which is similar to Mix$^+$ group. Meanwhile, the number of CFU-Cs from CD45$^+$ cells showed a similar trend in the Mix$^{+T}$ compared to Mix$^+$ group (Fig. 6N–P; Appendix Fig. S4F1). These data support that the contact between Mks (CD226$^+$ Mks) and HECs is dispensable for the hemogenic potential of HECs and Mks regulate EHT probably through secreted factors.

## Megakaryocytes promote the potential of hemogenic endothelial cells through TNFSF14/LTβR pathways

To further characterize how Mks promote the process of EHT, we analyzed the interaction between HECs and Mks by analyzing sc-RNA seq data from our Mks and Hou et al's HEC (Hou et al, 2020). The interaction number and strength of CD226$^+$ Mk-HEC were similar compared to that of CD226$^-$ Mk-HEC. Some pathways, such as *Periostin*, *Galectin*, *Tgfb*, *Bmp*, *Angpt*, *Fgf*, *Tnfsf14*, and *Tnf* were highly expressed in CD226$^+$ Mks, but the genes including *Mdk*, *Ptn*, *Sema3*, and *Wnt* were enriched in the CD226$^-$ Mks (Appendix Fig. S5A,B). The genes of ligand and receptor related to Mk-HEC interaction, for example, *Vegfc-Vegfr3*, *Tnfsf14* (LIGHT)-*Ltβr*, *Lgals9-Cd44*, *Cxcl5-Cxcr2*, *Angpt1-Tek* were enhanced obviously

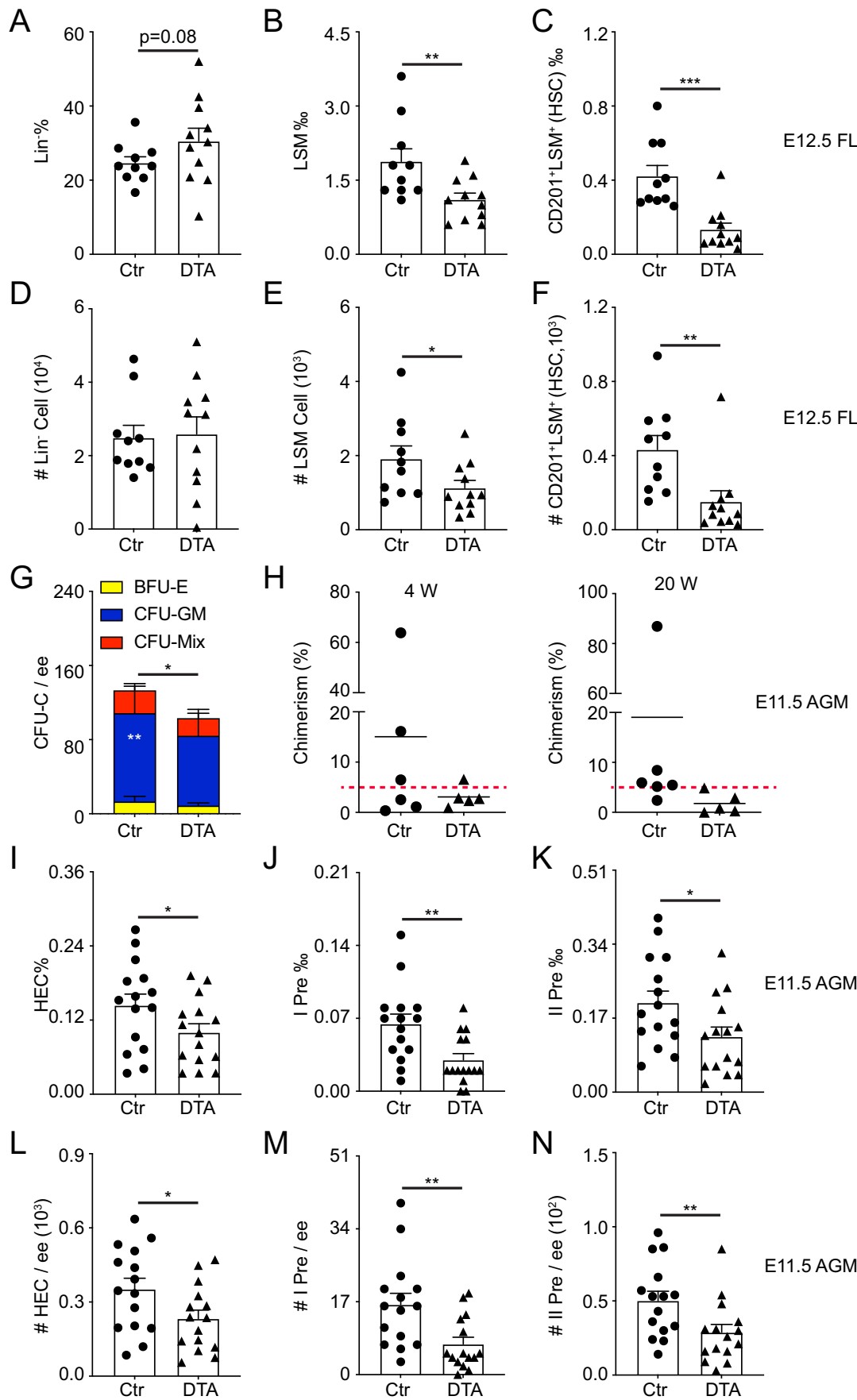

**Figure 3. Mk deletion leads to the reduction of HPCs and HSCs.**

(A–F) Flow cytometric analysis displaying the percentages of Lin⁻, Lin⁻Sca1⁺Mac1^low (LSM), and CD201⁺LSM (HSC) in E12.5 fetal liver cells. $n = 10$–11 fetuses from at least 3 litters/condition, $*p = 0.0298$, $**p < 0.01$, $***p = 0.0002$. Circle = control (Ctr), triangle = DTA. (G) Methylcellulose culture data showing the number of CFU-Cs and the number of each hematopoietic colony type (indicated by color bars) per embryo equivalent (ee) in the E11.5 AGM region. $n = 8$ independent experiments, $*p = 0.0112$, $**p = 0.0047$. (H) Percentage of donor cell (CD45.2/2) chimerism in peripheral blood of irradiated recipients receiving control or DTA AGM region cells with $2 \times 10^5$ bone marrow cells at transplantation 4 weeks and 20 weeks. Circles and triangles indicate individual recipients of control or DTA cells, respectively. $n = 5$–6 mice per each condition from independent 4 times experiments. Red dashed line represents 5% chimerism. (I–K) The percentage of hemogenic endothelial cells (HEC, CD41⁻CD45⁻CD31⁺CD44⁺CD201⁺c-Kit⁺) and pre-HSCs in the E11.5 control and PF4-Cre;Rosa-DTA (DTA) AGM region. $n = 15$ fetuses from at least 3 litters/condition, $*p < 0.05$, $**p = 0.0024$. (L–N) The reduction of cell number of HEC and pre-HSCs. $n = 15$ fetuses from at least 3 litters/condition, $*p = 0.0205$, $**p < 0.01$. Data Information: For all analysis above bars represent mean ± SEM. Statistical significance was determined by unpaired Student's t-test. Source data are available online for this figure.

between CD226⁺ Mks and HECs compared to the CD226⁻ Mks to HEC (Fig. 7A). Some of these genes (*Tnfsf14*, *Lgals9*, *Angpt1*, and so on) were expressed more highly in the CD226⁺ Mks from scRNA-seq data, however, qRT-PCR analysis confirmed that *Tnfsf14* was only one of these genes with higher expression in the CD226⁺ Mks compared to CD226⁻ Mks. *Lgals9* and *Angpt1* were expressed comparably in the CD226⁺ and CD226⁻ Mks fractions and the latter was highly expressed in the OP9-DL1 cells (Fig. 7B,C; Appendix Fig. S5C). Since our above data showed the regulation of Mks on the EHT process possibly was mediated by secreted factors, and the receptor of *Tnfsf14* (*Ltβr*) was expressed in the HEC fractions, although no differences between CD44⁺ EC (HEC) and CD44⁻ EC (non-HEC) (Appendix Fig. S5D). We focused on the TNFSF14 for further study because TNFSF14 is one kind of soluble factors.

It is reported that TNFSF14 regulates HS/PC differentiation (Hopner et al, 2021). We detected the roles of TNFSF14 in the EHT process, TNFSF14 promoted the emergence of CD45⁺ cells derived from HEC in the OP9-DL1 coculture system (Fig. 7D), which is a similar effect on the existence of Mk. Explant culture data showed that the number of CFU-Cs was reduced in the DTA group after 3 days explant cultures, in line with the reduction of pre-HSCs. Furthermore, the existence of TNFSF14 rescued the number of CFU-Cs in the DTA group (Fig. 7E; Appendix Fig. S3J,K). In addition, at 4 weeks transplantation, 2 out 4 recipients with chimerism 15.5 ± 8.3% were engrafted in the TNFSF14 treatment in DTA explant culture, similar to the control group (3/4 engrafted recipients with chimerism 11.3 ± 4.3%), however, one of four recipients (5.4 ± 3.7%) were positive in the DTA explant cultures. After 16 weeks transplantation, 3/4 recipients were reconstituted with chimerism 50.9 ± 19.5% in the control group. In the DTA+TNFSF14 group, one out of 3 recipients was engrafted with chimerism (87.9%) after 16 weeks transplantation, however, the chimerism of only one positive recipient (1/4) was 7.5% in the DTA group (Fig. 7F), indicating possible rescue function of TNFSF14 on DTA group. These data suggest that TNFSF14 derived from Mk enhances the hemopoietic ability in the embryos.

Since our previous study has shown that pro-inflammatory factors including TNF (TNFα, TNFSF1α) from Macs are able to promote the process of EHT and we have found *Tnf* is expressed in the CD226⁺ and CD226⁻ Mks (Appendix Fig. S5C). We wondered whether Mks and Macs play roles together in the EHT process. Comparison analysis from our single-cell RNA-sequencing data (Appendix Fig. S5E) (Liu et al, 2024) showed the different pathways of interaction between Mac-HEC and Mk-HEC. The former was enriched in the *Tnf-Tnfrsf1a*, *Angptl4-Cdh5*, and the latter was specifically enhanced for *Tnfsf14-Ltβr*, *Tgfb1-Tgfbr1 + 2*. Gene expression analysis confirmed the expression of *Tnfsf14* was still higher in the Mk cluster and *Tnf* in the Mac cluster (Appendix Fig. S5F–G). Our qRT-PCR data showed higher expression of *Tnf*

in the Mac of E10.5-E11.5 AGM regions (Appendix Fig. S5H), consistent with our previous studies (Li et al, 2019; Mariani et al, 2019). Then, the coculture system displayed that TNFSF14 along with TNF are more obvious enhancements of HPCs (CD45⁺ cells) production in the EHT process, in line with the increased number of CD45⁺ cells and CFU-Cs at the presence of Mac and/or Mk (Fig. 7G; Appendix Fig. S5I,J), indicating the roles of Mks and Macs during hematopoiesis in the embryo.

## Discussion

We have shown that Mks play a critical role in HSC development, especially in pre-HSC generation and maturation. They are involved in the earlier stage of EHT process through secreting TNFSF14. Although some pro-inflammatory factors derived from other cells, such as Macs, neutrophils were shown to affect HSC development in mouse and zebrafish embryos (Espin-Palazon et al, 2014; Mariani et al, 2019), it is the first time to demonstrate that Mks are important for hematopoiesis in the AGM region.

### PF4 specifically labels megakaryocytes in the AGM region

Although CXCL4 (PF4) inhibits the development of Mks, PF4 is a useful marker for enriching Mks in adults (Tiedt et al, 2007). The specificity of PF4 in Mks was detected by flow cytometric and immunostaining. Most PF4⁺ (tdTomato) cells (>90%) were co-expressed with CD41 or CD42d and only rare cells were positive for both CD45 and PF4, which might be relative to diploid platelet-forming cells (Potts et al, 2015; Potts et al, 2014) and in line with that CD45 coexpressed with some Mks (Cortegano et al, 2019; Lambert et al, 2009). Furthermore, almost none of hematopoietic-related cells including ECs and pre-HSCs were positive for PF4, indicating that PF4 specifically labels Mks at least in the detected stage.

A previous report has shown that PF4 is a negative autocrine in megakaryopoiesis (Lambert et al, 2007). However, one publication has shown that early Mk development roughly appears normal in the PF4-Cre;Rosa-DTA embryos (E11.5-E15.5) depending on the morphology observation (Carramolino et al, 2010). Based on more accurate flow cytometric analysis, the reduction of Mks was observed in the E10.5-E11.5 AGM regions as well as in the fetal liver, in line with previous reports (Carramolino et al, 2010). Consistently, the ability of Mk maturation and platelet production was lessened in the DTA yolk sac. Thus, PF4-Cre;Rosa-DTA double transgenic mouse model is the affable model to study the role of Mks in the embryos, although this mouse model fails to affect MkPs

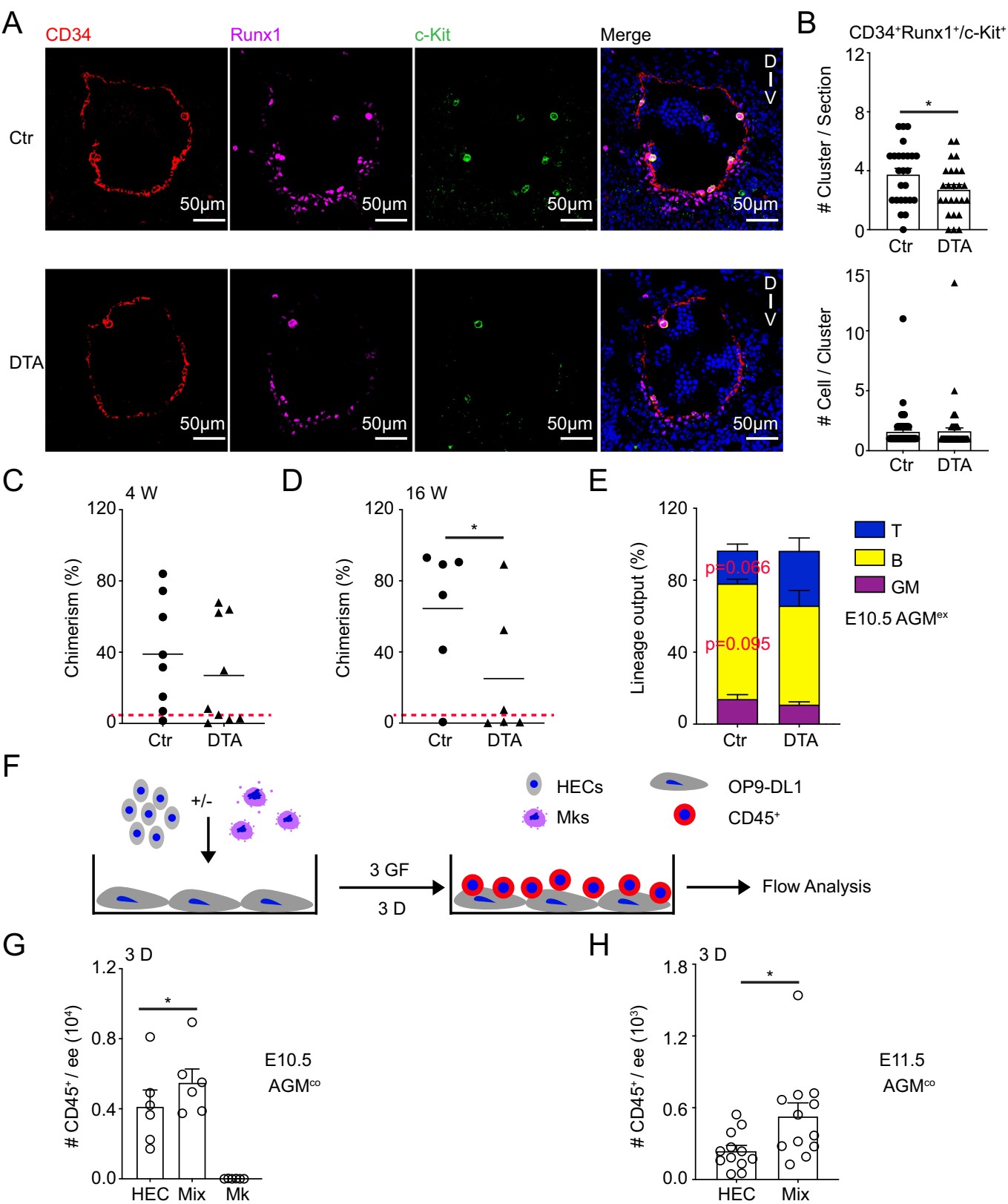

**Figure 4.  The roles of Mks in the process of EHT.**

(A) Immunostaining of cryosections displaying hematopoietic clusters (defined by CD34⁺Runx1⁺ and CD34⁺c-Kit⁺) in the E10.5 embryos. Scale Bar presents 50 μm. Red = CD34, Megenta = Runx1, Green = c-Kit, Blue = Hoechst. (B) The decreased number of hematopoietic clusters (CD34⁺Runx1⁺ and CD34⁺c-Kit⁺) per section and unchanged cell numbers of each hematopoietic clusters in the E10.5 DTA AGM region. $n = 3$ fetuses from 3 litters/condition. Statistical significance was determined by Mann–Whitney test, $^*p = 0.0376$. (C, D) Percentages of donor cell (CD45.2/2) chimerism in peripheral blood of irradiated recipients receiving control or DTA AGMᵉˣ cells after transplantation 4 or 16 weeks. Circles and triangles indicate individual recipients of control (Ctr) or DTA cells, respectively. $n = 6–9$ mice per each condition from 5 independent experiments, $^*p = 0.0479$. Red dashed line represents 5% chimerism. (E) The myeloid and lymphoid lineage output in the recipients receiving control and DTA AGMᵉˣ cells. $n = 3–5$ mice per each condition from 3 independent experiments. Red dashed line represents 5% chimerism. (F) The schematic of OP9-DL1 cocultures with endothelial cells and Mks. (G) CD45⁺ hematopoietic cells derived from E10.5 AGM hemogenic endothelial cells ± Mks by OP9-DL1 coculture system for 3 days. $n = 6$ independent experiments. Statistical significance was determined by paired Student's t-test, $^*p = 0.0382$. (H) The roles of Mks in promoting endothelial to hematopoietic cells (CD45⁺) transition after 3 days OP9-DL1 coculture in the E11.5 AGM region. $n = 12$ independent experiments. Statistical significance was determined by paired Student's t-test. $^*p = 0.0248$. HEC = hemogenic endothelial cells (CD41⁻CD45⁻CD31⁺CD44⁺), Mix = HEC+Mks, Mks = megakaryocytes. 3 D = 3 days. Data Information: For all analysis above bars represent mean ± SEM. Statistical significance was determined by unpaired Student's t-test unless the statistical test was indicated. Source data are available online for this figure.

and Mks development in the earlier stage and couldn't distinguish the developmental dynamics of MkP-Mk-platelets.

## Megakaryocytes regulate HSC maturation/differentiation in the AGM region

Distinct groups have reported that Mks play roles in both maintaining HSC quiescence during homeostasis and promoting HSC regeneration after chemotherapeutic stress, through TGF-ß1 and FGF1, respectively (Bruns et al, 2014; Zhao et al, 2014). Mk depletion reprograms vWF⁺ HSC from myeloid-biased to balanced-lineage (Pinho et al, 2018). In the fetal liver, phenotypically defined HSC was reduced when Mks were diminished, consistent with the situation in adults. In the stage of HSC emergence, Mk reduction induced the decreased trend of HSC activity. Moreover, the detection of HSC precursors by explant cultures confirmed the decrease in engraftment. In addition, in vitro cultures showed the possible disrupted pre-HSC maturation or differentiation in the Mk-depleted group, consistent with that Mks modify HSC fate in the bone marrow (Pinho et al, 2018). Taken together, Mks regulate the HSC maturation/differentiation in the AGM region.

## Megakaryocytes regulate the EHT process in vitro

Abundant factors play roles in the EHT process in the embryos (Dzierzak and Bigas, 2018). Niche cells of AGM region contribute distinct factors in the regulation of HSC emergence, such as the mesenchymal cell for BMP4 and Macs for TNF (Li et al, 2019; Mariani et al, 2019). It is unknown the function of Mks on hematopoietic cell generation in the earlier stage since Mks emerge as early as erythroid cells in the yolk sac. Depending on the deletion model of Mks, the development of pre-HSCs into functional HSCs was blocked partially. We found that Mks influenced pre-HSC production/maturation only in the late stage (E11.5) but not in E10.5. However, our coculture results displayed that Mks boosted the hemogenic potential of HECs after 3 days coculture. Since the reduction of Mks started to be detected in the E10.5 AGM region of PF4-Cre;Rosa-DTA mouse model, it takes time for the physiological functions of Mks on HSC activity and the roles might be underestimated. That is the partial reason for the disabled alteration of hematopoietic cells in the E10.5 AGM region. Whatever, our data indeed support the physiological role of Mks in hematopoiesis in the embryos.

## Pro-inflammatory factors secreted by mature megakaryocytes (CD226⁺ Mks) are involved in hematopoiesis

The heterogeneity of Mks is indicated by serval groups, including immune-biased Mks. And abundance markers are found for separating subsets of Mks (Sun et al, 2021; Wang et al, 2021). CD226 (DNAX accessory molecule-1 (DNAM-1)) is an immunoglobulin-like glycoprotein and mainly identified as a co-stimulatory immune receptor on T and NK cells, which has a crucial function in immunology (Iguchi-Manaka et al, 2008; Shibuya et al, 1996; Wagner et al, 2017). CD226 also regulates the migration of monocytes through endothelial junctions (Reymond et al, 2004). Previous studies have shown high expression of CD226 on the Mks, affecting the ploidy of Mks (Ma et al, 2005). The deletion of CD226 specifically on Mks disrupts the balance between Mks and platelets and activates the platelets (Bian et al, 2020; Sun et al, 2021). Consistently, CD226 was selected from our scRNA-seq data for separating the subset of Mks (mature Mks). In addition, it is reported that CD226 plays an important role in adhesive interaction between ECs and Mks and mitochondria damage of ECs (Kojima et al, 2003; Zhou et al, 2021). Our discrimination of Mks supports that mature Mk plays a positive role in the EHT process, suggesting that Mks are essential for EHT.

TGF-β1 and FGF1 derived from Mks regulate HSC quiescence in bone marrow (Zhao et al, 2014), indicating that contact between Mks and HSCs is not necessary. From our data, cocultures with transwell showed that interaction between Mks and ECs is dispensable for the promotion of EHT. Comparison bioinformatics analysis between CD226⁺ and CD226⁻ cluster cells and qRT-PCR confirmed the higher expression of *Tnfsf14* in the CD226⁺ Mks, which is in line with previous studies that TNFSF14 is associated with platelet activation (Chen et al, 2018).

TNFSF14 (LIGHT) is a secreted protein of the TNF superfamily, which is the ligand for LTβR and herpes virus entry mediator (HVEM). LTβR is mainly expressed in ECs and HVEM in lymphoid cells (Sedy et al, 2014; Ware et al, 2022). Consistently, LTβR is expressed in the ECs (HECs) of AGM region. Furthermore, cell chat analysis from Mks and HECs has shown that *Tnfsf14-Ltβr* pathways are strong expressions in the CD226⁺ Mk clusters. A previous study has shown that the upregulation of TNFSF14 is related to HPC proliferation and differentiation (Chen et al, 2018). TNFSF14/LTβR maintains the HSC pool through control symmetric division (Hopner et al, 2021). Although the concentration of TNFSF14 protein secreted from Mks is too low to be detected by

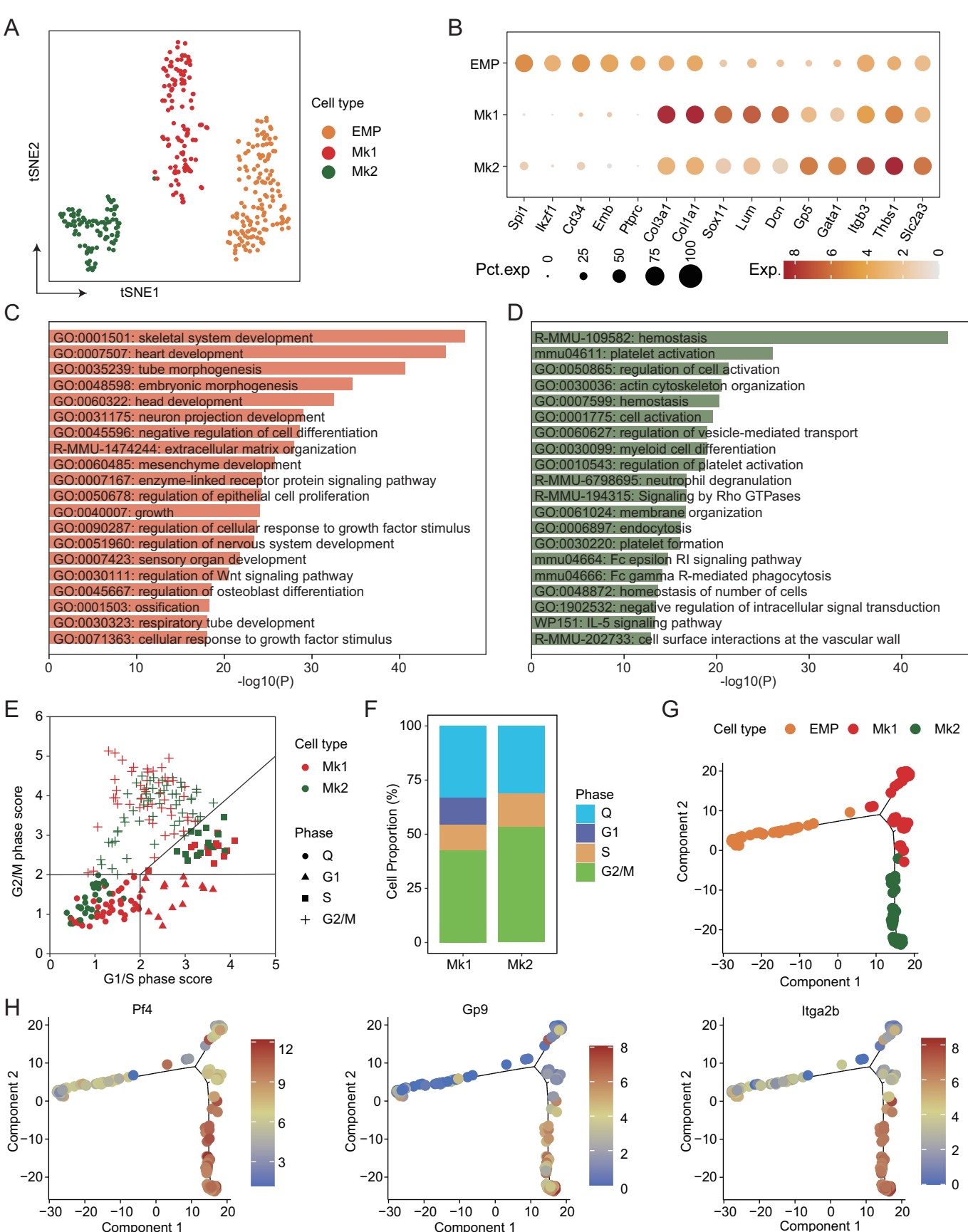

◄ **Figure 5. Single-cell RNA-sequencing analysis identifying the heterogeneity of Mks.**

(A) TSNE plot visualized three clusters from flow cytometric sorted CD41^low c-Kit^+ EMPs and CD41^+CD42d^+ Mks. (B) Featured gene expression in the EMP and Mk clusters. (C, D) Metascape analysis displaying the enrichment of gene function in the Mk1 (C) and Mk2 (D). Statistical significance was determined by hypergeometric test and adjusted by Benjamini-Hochberg p-value correction algorithm. (E) Cell cycle score analysis showing the differences in cell cycle status between Mk1 and Mk2. (F) The percentage contribution of different cell cycle phases in the Mk1 and Mk2 cells. (G) Pseudotime differences by trajectory analysis in the EMP, Mk1, and Mk2. (H) The expression of *Pf4*, *Gp9*, and *Itga2b* in the EMP, Mk1, and Mk2 according to pseudotime analysis. EMP = erythroid-myeloid progenitors. Mk = Megakaryocytes.

the commercial reagent because of the lower number of CD226^+ Mks, recombinant TNFSF14 promotes the EHT process by coculture system and TNFSF14 is able to rescue the ability of hematopoiesis (HPCs, but not HSC because of the limited recipient in the rescue group) in the Mk deficiency explant culture system. In addition, recent reports have shown that PF4 is involved in neurogenesis and cognition ability (Leiter et al, 2023; Park et al, 2023). Probably, PF4 derived from Mks plays a role in hematopoietic development, which needs to be investigated further.

Our previous studies have shown the regulatory roles of TNF derived from Macs (Li et al, 2019; Mariani et al, 2019). And TNFSF14 secreted by Mks is also involved in the hematopoietic cell emergence in this study. The regulatory mechanism of TNFSF14 and TNF might go through distinct downstream pathways (non-canonical or canonical NFκB signaling pathways, respectively) (Piao et al, 2021). Taken together, Mks play an important role in hematopoietic development, especially in the emergence of hematopoietic precursors, likely via secreting TNFSF14, although the origin of Mks remains to be addressed.

# Methods

## Mouse and embryo generation

PF4-Cre (Tiedt et al, 2007) (from Meng Zhao), Rosa-tdTomato (Madisen et al, 2010) (from Bing Liu), Rosa-DTA (The Jackson Laboratory, JAX: 009669) mice were used for timed mating and C57BL/6-Ly5.1/1 (Janowska-Wieczorek et al, 2001) mice (8–12 weeks) were as transplantation recipients. PF4-Cre;Rosa-tdTomato embryos were generated from PF4-Cre male mice crossed with Rosa-tdTomato female mice. Male PF4-Cre mice were crossed with female Rosa-DTA mice for PF4-Cre;Rosa-DTA embryos (Ly5.2/2). In embryos, AGM region, yolk sac (excluding the vitellin/umbilical vessels), and fetal liver were dissected. The embryo stages were identified by counting somite pairs and tails were used for genotyping.

Mice were housed in the animal facility of Southern Medical University and mice experiments were approved by the ethics committee of Southern Medical University.

## Hematopoietic progenitor and stem cell assays

Cell suspensions from AGM and fetal liver or cultures were cultured in the methylcellulose (M3434; Stem Cell Technologies) for CFU-C assay. The counting and quantification of CFU-C was according to previous report (Li et al, 2019). E11.5 AGM cells or AGM explant (AGM^ex) cells (1 ee, Ly5.2/2) were injected intravenously with supporting bone marrow cells (2 × 10^5 leukocytes Ly5.1/1) into 8.5 Gy (4.5 Gy+4 Gy) irradiated recipient mice

(Ly5.1/1). Chimerism and lineage output assays were performed at 4, 16/20 weeks after transplantation from peripheral blood.

## Explant cultures and OP9-DL1 cocultures

AGM explants (AGM^ex) were performed as previously described (Medvinsky and Dzierzak, 1996; Muller et al, 1994). Briefly, AGM regions were laid on a nylon membrane (Millipore) with metallic supports and cultured in MyeloCult M5300 or H5100 (Stem Cell Technologies) supplemented with 10 μM hydrocortisone (Sigma-Aldrich). After 3 days culture, explants were digested into single cells by 0.125% collagenase digestion (Sigma-Aldrich) for flow cytometry analysis or further culture. In some conditions, TNFSF14 or TNF was added into the explant cultures.

Flow-sorted ECs, pre-HSC I and pre-HSC II cells were cocultured with OP9-DL1 cells (stem cell factor, 100 ng/mL; IL-3, 100 ng/mL; and Flt3-ligand, 100 ng/mL; PeproTech) or OP9 cells (stem cell factor, 20 ng/mL; IL-7, 10 ng/mL; and Flt3-ligand, 10 ng/mL; PeproTech) for 3–10 days as previous report (Li et al, 2013). In some conditions, transwells were used to separate Mks and HECs. Cells were harvested by mechanical pipetting for flow cytometric analysis.

## Flow cytometric analysis

Cells from AGM regions and fetal liver were used for flow cytometric analysis. In the AGM region, cells were prepared for staining by different antibodies: CD31, CD41, CD45, CD44, CD201, c-Kit for HECs, and pre-HSCs. 7AAD or Hoechst was used for excluding dead cells. In fetal liver, lineage cocktail (Ter119, Gr1, NK1.1, CD3, B220), Sca1, c-Kit, CD150, CD48, Mac1 and CD201 were used for separating HS/PCs (Lin^-Sca1^+Mac1^low, or Lin^-Sca1^+c-Kit^+) and HSCs (Lin^-Sca1^+Mac1^lowCD201^+, or Lin^-Sca1^+c-Kit^+CD150^+CD48^-) (Zhou et al, 2016). Antibodies were listed in the Dataset EV3.

## Immunostaining

Fluorescence immunostaining was performed as described previously (Hou et al, 2020). E10.5 embryos were fixed (2% paraformaldehyde/PBS, 20 min, 4 °C), equilibrated in 20% sucrose/PBS at 4 °C overnight, and then embedded in the Tissue Tek before freezing. In each control and DTA embryo, we sequenced the sections and chose 4–5 sections per embryo from ~100 sections at similar areas along the rostral-caudal axis. After additional blocking of the endogenous biotin step, antibody staining was performed. Primary and secondary antibodies were used at the following concentrations: Anti-c-Kit (1:100, BD), biotinylated anti-CD34 (1:200, eBioscience), Runx1 (1:500, Abcam), CD41 (1:200), tdTomato (1:200) and Hoechst for nuclear staining. APC-CD226 (1:100) was used for staining. In addition,

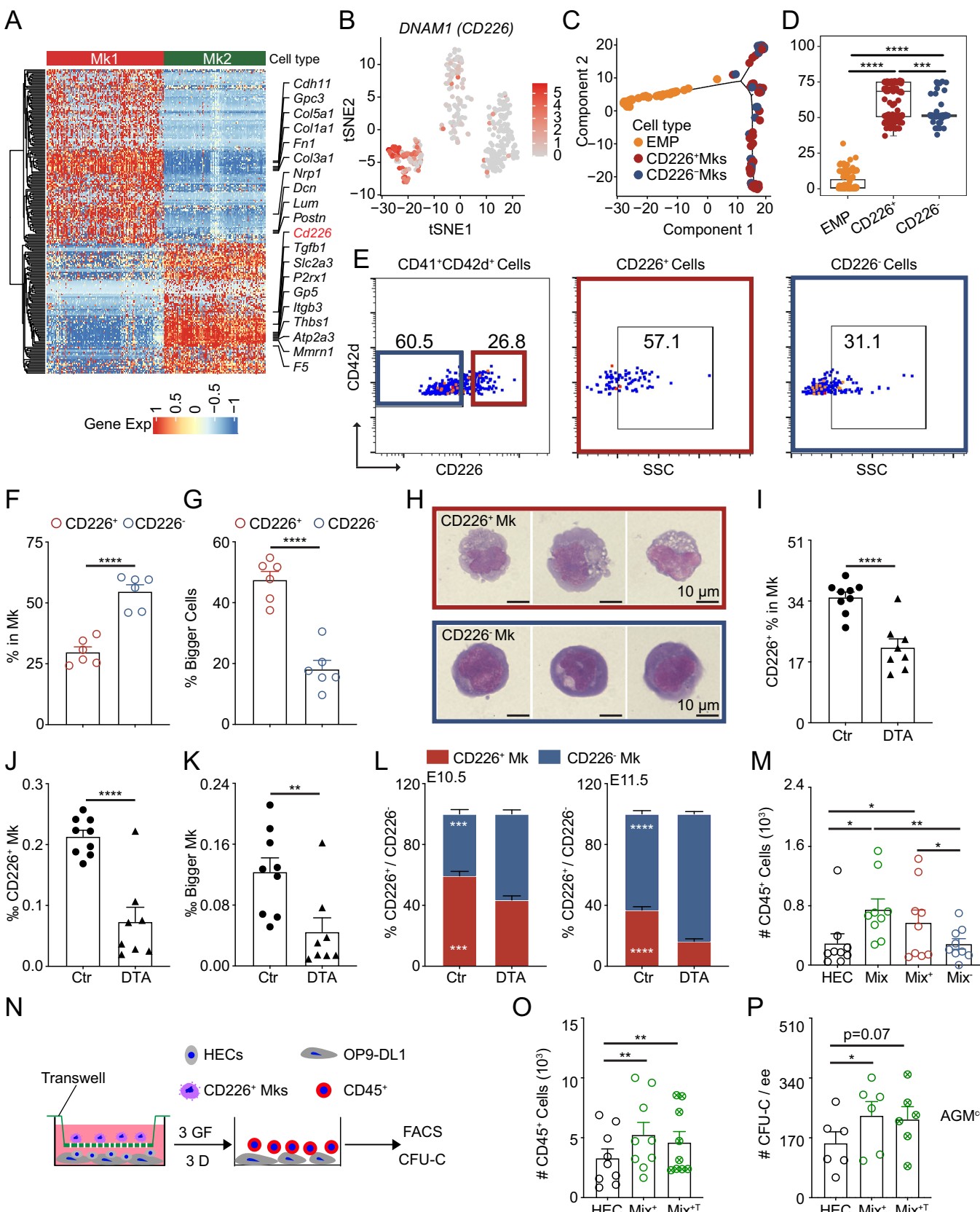

**Figure 6. The subfraction of Mks discriminated by CD226 affects EHT.**

(A) Different gene expression of surface markers between Mk1 and Mk2. (B) The higher expression of *DNAM1* (*CD226*) in the tSNE graph. (C, D) The trajectory analysis showing CD226+ cluster cells in the late stage compared with CD226- cluster cells. The box plot's central band marks the median, boxes mark the first and third quartiles, and whiskers extend the boxes to the largest value no further than 1.5 times the interquartile range. $n = 3$ independent experiments. Statistical significance was determined by Wilcoxon test. ***$p = 0.00022$, ****$p < 0.0001$. (E) Presentative flow cytometric analysis showing the enrichment of CD226 in Mks (CD41+CD42d+). (F) The percentages of CD226+ and CD226- cells in the Mks of E11.5 AGM region. $n = 6$ fetuses from 6 litters/condition, ****$p < 0.0001$. (G) The percentages of bigger size cells (SSH^high) in the CD226+ and CD226- fraction of Mks. $n = 6$ fetuses from 6 litters/condition, ****$p < 0.0001$. (H) The morphology in the E11.5 AGM region displaying higher polyploid and more granules in the CD226+ Mks compared with CD226- Mks. The scale bar presents 10 μm. (I) Percentages of CD226+ in Mks in the E11.5 control and DTA AGM region. $n = 8$–9 fetuses from at least 3 litters/condition, ****$p < 0.0001$. (J, K) The decrease of CD226+ Mks (J) and Bigger Mks (K) in the E11.5 DTA AGM region compared to control. $n = 8$–9 fetuses from at least 3 litters/condition, **$p = 0.0043$, ****$p < 0.0001$. (L) The ratios between CD226+ and CD226- percentage in the E10.5-E11.5 control and DTA AGM region. $n = 8$–10 fetuses from at least 3 litters/condition, ***$p < 0.001$, ****$p < 0.0001$. (M) Bar graph showing that the existence of CD226+ Mks enhances the production of hematopoietic cells from endothelial cells in the E11.5 AGM region. HEC = hemogenic endothelial cells (CD41-CD45-CD31+CD44+), Mix = HEC+Mks, Mix+ = HEC + CD226+ Mks, Mix- = HEC + CD226- Mks. $n = 9$ independent experiments. Statistical significance was determined by paired Student's t-test, *$p < 0.05$, **$p = 0.0099$. (N) The schematic of OP9-DL1 cocultures with or without transwells. (O) Transwell coculture assay indicating that the interaction between CD226+ Mks and HEC failed to affect the emergence of CD45+ cells. $n = 9$ independent experiments. Statistical significance was determined by paired Student's t-test, **$p < 0.01$. (P) CFU-Cs from CD45+ cells in the OP9-DL1 coculture system. HEC=hemogenic endothelial cells, Mix+ = HEC + CD226+Mks, Mix+T = HEC + CD226+Mks +Transwells. $n = 6$ independent experiments. Statistical significance was determined by paired Student's t-test, *$p = 0.0331$. Data Information: For all analysis above bars represent mean ± SEM. Statistical significance was determined by unpaired Student's t-test unless the statistical test was indicated. Source data are available online for this figure.

CD41 (1:100) was used for staining cocultures from yolk sac. Secondary antibodies (Invitrogen): Alexa Flour 647 anti-rat IgG (1:1000), Alexa Flour 488 anti-rabbit IgG (1:1000), and streptavidin Cy3 (1:1000), streptavidin Alexa Fluor 488 (1:1000). The image procedures were performed by confocal microscope (Zeiss LSM 880).

For counting the number of Mks in the distinct location of the DA, the areas were circled between the distances 10, 50, and 150 μm outside of DA based on the DA endothelial border (depending on the staining of CD34) by Image J. The numbers of Mks and platelets were counted based on the expression of CD41 or tdTamato and Image J was used for calculating the square areas.

## Gene expression analysis

RNA from sorted cells was extracted and complementary DNAs were generated with oligdTprimers and SuperScriptII Reverse Transcriptase (Thermo Fisher Scientific). Real-time (RT) PCR was performed by using FastSYBR Green Master Mix and ABI7900 (Thermo Fisher Scientific) detection. Primers are listed in Appendix Table S1.

## Single-cell RNA sequencing

For well-based scRNA-seq (modified STRT-seq), AGM Mks were sorted by flow cytometry according to Ter119-CD45-CD41+CD42d+cells (Appendix Fig. S4A). Libraries were produced following the previous report (Hou et al, 2020; Li et al, 2017a; Picelli et al, 2013; Picelli et al, 2014) and sequenced on Illumina Novaseq 6000 platform in a 150 bp paired-end manner (sequenced by Berry Genomics).

## Processing and quality control of scRNA-seq raw data

For modified STRT-seq data, we used UMI-based scRNA-seq method to accurately measure the gene expression profiles within individual cells: (1) Distributing the raw data into each cell according to the barcode in the original sequencing data read 2; The corresponding read 1 was trimmed to remove the TSO sequence and polyA tail sequence after UMI information was aligned to it; (2) Using the software Hisat2 (v 2.1.0) to align read 1 with the mouse reference genome GRCm39; (3) Transcript count for each gene using htseq-count (v 0.11.3) and defined as a unique molecular identifier (UMI) for gene expression in each cell; (4) Low-quality cells were removed if they met any of the following criteria: (1) <10,000 unique molecular identifiers (UMI); (2) <2000 genes; (3) >10% UMIs derived from the mitochondrial genome; (4) <20% a mapping ratio of reference genome; (5) 351 cells were obtained. Gene expression levels are normalized by $\log_2$ (transcript per million (TPM)/10 + 1) (Dataset EV1).

## Seurat analysis procedure for scRNA-seq datasets

Seurat (v 4.3.0) was employed for advanced analysis of STRT-seq datasets. For different STRT-seq datasets, we followed similar Seurat analysis procedure. Briefly, we first used "NormalizeData", "FindVariableFeatures" and "ScaleData" functions to perform the workflow of data normalization, identification of highly variable genes (HVGs), and then used "RunPCA" function to perform PCA dimensionality reduction. Finally, The data were used in the "RunTSNE" function and cluster analysis by the "FindNeighbors" and "FindClusters" functions.

## Cluster feature genes identification

After the projection of all cells into two-dimensional spaces by tSNE, cells were clustered according to common features. Feature genes of each cell cluster were identified with the "FindAllMarkers" function in Seurat. We set the parameters "only.pos" as "TRUE" and "min.pct = 0.2", which only returned the genes expressed at a significantly higher level in a given cluster. By using the Wilcoxon test, log2 foldchange >1 and adjusted $p$ value < 0.05 was used for screening feature genes (Dataset EV1).

## Gene enrichment analysis

We used Metascape (v 3.5.20230501) for gene set enrichment analysis and Gene ontology (GO) analysis was performed ClueGO (v 2.5.8) plugin in Cytoscape (v 3.9.1).

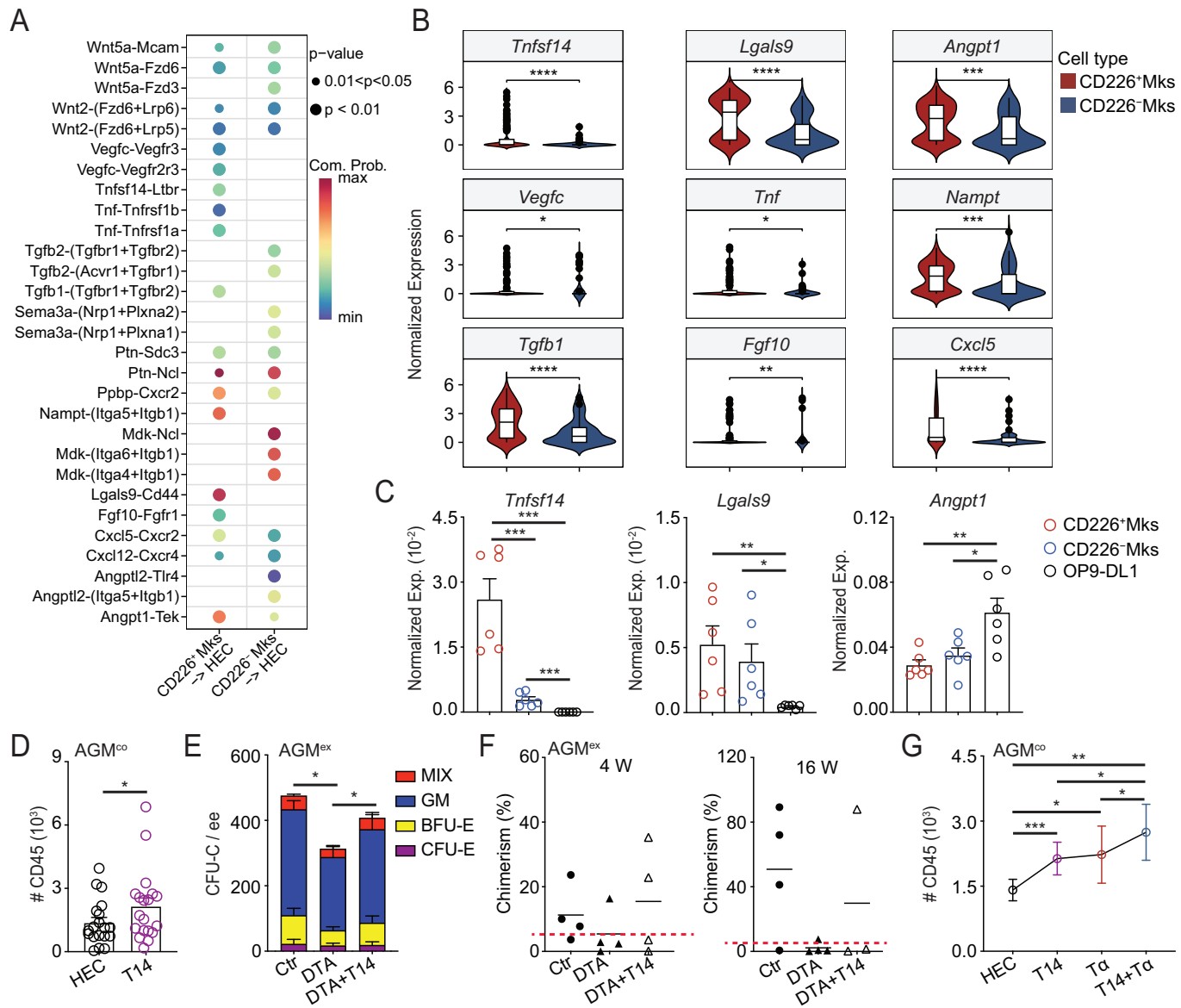

**Figure 7. TNFSF14 secreted by CD226⁺ Mks boosts the hemogenic potential of HECs.**

(A) Comparative analysis showing the interactions between CD226⁺/CD226⁻ Mks and HECs based on the ligands-receptors. Statistical significance was determined by one-sided permutation test. (B) Transcriptomic gene expression in the CD226⁺ and CD226⁻ cluster cells. The box plot's central band marks the median, boxes mark the first and third quartiles. $n = 3$ independent experiments. Statistical significance was determined by Wilcoxon test, $*p < 0.05$, $**p < 0.01$, $***p < 0.001$, $****p < 0.0001$. (C) qRT-PCR confirmed the expression of selected genes in the CD226⁺ Mks and CD226⁻ Mks as well as OP9-DL1. $n = 3$ independent experiments, $*p < 0.05$, $**p < 0.01$, $***p < 0.001$. (D) The enhancement of CD45⁺ hematopoietic cells from hemogenic endothelial cells (HEC, CD41⁻CD45⁻CD31⁺CD44⁺) by TNFSF14 treatment in the OP9-DL1 coculture system. $n = 19$ independent experiments. Statistical significance was determined by paired Student's t-test, $*p = 0.048$. T14 = TNFSF14. (E) Methylcellulose analysis showing the rescue function of TNFSF14 in the number of hemopoietic progenitor cells after 3 days AGM (DTA) explant culture. $n = 4$ independent experiments, $*p < 0.05$. T14 = TNFSF14. (F) Percentages of donor cell (CD45.2/2) chimerism in peripheral blood of irradiated recipients receiving control or DTA AGMᵉˣ cells after transplantation 4 or 16 weeks with or without TNFSF14. Circles = Control, black solid triangles = DTA cells and black hollow triangles = DTA cells with TNFSF14 treatment (DTA + T14). $n = 3$–4 mice per each condition from 3 independent experiments. T14 = TNFSF14. Red dashed line represents 5% chimerism. (G) The hemogenic capacity of HECs was boosted in the existence of both TNF and TNFSF14 compared with the treatment by each of them. $n \geq 8$ independent experiments. Statistical significance was determined by paired Student's t-test, $*p < 0.05$, $**p = 0.0019$, $***p = 0.0008$. T14 = TNFSF14, Tα = TNFα. Data Information: For all analysis above bars represent mean ± SEM. Statistical significance was determined by unpaired Student's t-test unless the statistical test was indicated. Source data are available online for this figure.

## Cell cycle analysis

We used "CellCycleScoring" function in Seurat to calculate the cell cycle phase scores and achieve cell cycle phase assignments for single cells. Cell cycle-related genes including a previously defined core set of 43 G1/S and 54 G2/M genes (Macosko et al, 2015), were used for cell cycle analysis (Dataset EV1) (Hou et al, 2022).

## Trajectory analysis

In order to determine the developmental trajectory of Mks, as well as the maturation status of Mks, the Monocle 2 package (v 2.22.0) was used to reconstruct pseudotime trajectories according to the tutorials (http://cole-trapnell-lab.github.io/monocle-release/tutorials/). Based on the "differentialGeneTest" function, the differentially expressed genes (q-value < 0.01) were selected as ordering genes. Then, the discriminative dimensionality reduction with trees (DDRTree) method was used for dimensionality reduction, which was visualized through the "plot_cell_trajectory" function.

## Identify subsets of megakaryocytes

We defined the *Dnam1* (*Cd226*) gene expression value (log$_2$ (TPM/10 + 1)) > 0 as CD226$^+$ Mk cluster cells and this value ≤0 as CD226$^-$ Mk cluster cells.

## Identification and analysis of differentially expressed genes (DEGs)

Differentially expressed genes between CD226$^+$ Mk cluster cells and CD226$^-$ Mk cluster cells were identified by "FindMarkers" function in Seurat. By using the Wilcoxon test, log2 foldchange > 0.8 and adjusted *p*-value < 0.05 were considered the screening conditions for the DEGs.

## Cell-cell interactions analysis

Ligand–receptor interactions were identified by CellChat (v 1.4.0) with default parameters. Only ligand–receptor interaction pairs with *p*-value < 0.05 returned by CellChat were considered significant. At the same time, CellChat can quantitatively infer and analyze intercellular communication networks. Cellular communication analysis using an officially recommended tutorial (https://github.com/sqjin/CellChat/blob/master/tutorial/).

## Identifying differences of TFs and cell surface markers

According to 1536 TF in AnimalTFDB3.0, TFs were selected from feature genes, and surface markers were marked based on 1296 validated surfaceome proteins identified in Cell Surface Protein Atlas. By using the Wilcoxon test, log2 foldchange >1 and adjusted *p*-value < 0.01 was considered the screening criteria for different expressed TFs and cell surface markers (Dataset EV2, EV4).

## Prediction of regulon activity scores and map TF target gene regulatory networks

Prediction of regulon activity scores was performed using the pyscenic CLI pipeline with default parameters, respectively (https://github.com/aertslab/pySCENIC). And using Cytoscape (v 3.9.1) to map the TF target gene regulatory network (Dataset EV2).

## Quantification and statistical analysis

All graphs were generated using GraphPad Prism 8. All data are presented as the mean ± SEM. Student t-test is for comparisons of 2 groups except for immunostaining data. and one-way ANOVA analysis of variance test is for comparisons of >2 groups. $P < 0.05$ was considered significant. *$p < 0.05$, **$p < 0.01$, ***$p < 0.001$, ****$p < 0.0001$. Further statistical details of experiments can be found in the figure legends. The number of biological replicates is indicated with '*n*'.

## Data availability

The raw sequencing data generated in the present study are deposited in the Genome Sequence Archive (Chen et al, 2021) in National Genomics Data Center (Members C-N, Partners 2023), China National Center for Bioinformation/Beijing Institute of Genomics, Chinese Academy of Sciences (https://ngdc.cncb.ac.cn/gsa, GSA: CRA011534). The processed datasets generated in this study are deposited in the OMIX, China National Center for Bioinformation/Beijing Institute of Genomics, Chinese Academy of Sciences (https://ngdc.cncb.ac.cn/omix, OMIX: OMIX004868).

## Peer review information

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

## Acknowledgements

The authors thank all lab members for their helpful comments. We also thank Dr. Meng Zhao for kind providing PF4-Cre mice and Rosa-DTA mice and Dr. Bing Liu for tdTomato mice. We thank Dr. Bing Liu, and Dr. Yu Lan for the critical discussion. The authors thank the Institute of Hematology, Jinan University, for providing flow cytometric sorting. This work was supported by grants from the National Key Research and Development Program (2019YFA0801802, 2019YFA0111100), and the National Natural Science Foundation of China (82270119).

## Author contributions

**Wenlang Lan**: Conceptualization; Data curation; Formal analysis; Investigation; Methodology; Writing—review and editing. **Jinping Li**: Conceptualization; Data curation; Formal analysis; Investigation; Methodology; Writing—review and editing. **Zehua Ye**: Conceptualization; Data curation; Formal analysis; Investigation; Writing—review and editing. **Yumin Liu**: Formal analysis. **Sifan Luo**: Methodology. **Xun Lu**: Formal analysis. **Zhan Cao**: Methodology. **Yifan Chen**: Formal analysis. **Hongtian Chen**: Methodology. **Zhuan Li**: Conceptualization; Resources; Data curation; Formal analysis; Supervision; Funding acquisition; Writing—original draft; Project administration; Writing—review and editing.

## Disclosure and competing interests statement

The authors declare no competing interests.

