## [Peer Review File · The EMBO Journal]

A subset of Megakaryocytes Regulates Development of Hematopoietic Stem Cell Precursors

Zhuan Li, Wenlang Lan, Jinping Li, Zehua Ye, Yumin Liu, Sifan Luo, Xun Lu, Zhan Cao, Yifan Chen, and Hongtian Chen

Corresponding author(s): Zhuan Li (zhuanli2018@smu.edu.cn)

Review Timeline:

Submission Date:	8th Sep 23
Editorial Decision:	27th Oct 23
Revision Received:	16th Jan 24
Editorial Decision:	21st Feb 24
Revision Received:	26th Feb 24
Accepted:	28th Feb 24

Editor: Daniel Klimmeck

Transaction Report:

Dear Dr Li,

Thank you again for submitting your manuscript EMBOJ-2023-115554 for consideration by the EMBO Journal. Please accept my sincere apologies for getting back to you with this unusual protraction due to delayed referee input, as well as detailed discussion in the editorial team. As indicated, your manuscript has been seen by three referees with expertise in hematopoietic stem cell biology and development, and we have received reports from all of them, which are shown below.

As you will see from their comments, the referees acknowledge the potential interest and value of your findings for the field. However, they also express major concerns, which need to be addressed thoroughly to make them supportive of publication in the EMBO Journal. In more detail, the referees #1 and #3 point to substantial limitations and concerns regarding comprehensive support on the in vivo claims presented (ref#1, pt.1; ref#3). Reviewer #2 in addition is concerned that the Pf4 reporter data might be confounded by additional expressing cell populations and requests important controls related (ref#2, pts.1,2). In addition, the reviewers raise a number of points related to additional controls required, overall data discussion and literature references that would need to be conclusively addressed to achieve the level of robustness and clarity needed for The EMBO Journal.

Given the overall interest stated and broader angle of your findings, we are able to invite you to revise your manuscript experimentally to address the referees' comments. I need to stress though that we do require strong support from the referees on a revised version of the study in order to move on to publication of the work.

In light of the extensive experimentation requested, I would appreciate if you could contact me during the next weeks for exchange e.g. a video call to discuss your perspective on the comments and potential plan for revisions.

Please feel free to contact me if you have any questions or need further input on the referee comments.

When submitting your revised manuscript, please carefully review the instructions below.

Please feel free to approach me any time should you have additional questions related to this.

Thank you for the opportunity to consider your work for publication.

I look forward to your revision.

Kind regards,

Daniel Klimmeck

Daniel Klimmeck, PhD
Senior Editor
The EMBO Journal

Instruction for the preparation of your revised manuscript:

- 1) a .docx formatted version of the manuscript text (including legends for main figures, EV figures and tables). Please make sure that the changes are highlighted to be clearly visible.
- 2) individual production quality figure files as .eps, .tif, .jpg (one file per figure).
- 3) a .docx formatted letter INCLUDING the reviewers' reports and your detailed point-by-point response to their comments. As part of the EMBO Press transparent editorial process, the point-by-point response is part of the Review Process File (RPF), which will be published alongside your paper.

4) a complete author checklist, which you can download from our author guidelines ([https://wol-prod-cdn.literatumonline.com/pb-assets/embo-site/Author Checklist%20-%20EMBO%20J-1561436015657.xlsx](https://wol-prod-cdn.literatumonline.com/pb-assets/embo-site/Author%20Checklist%20-%20EMBO%20J-1561436015657.xlsx)). Please insert information in the checklist that is also reflected in the manuscript. The completed author checklist will also be part of the RPF.

6) It is mandatory to include a 'Data Availability' section after the Materials and Methods. Before submitting your revision, primary datasets produced in this study need to be deposited in an appropriate public database, and the accession numbers and database listed under 'Data Availability'. Please remember to provide a reviewer password if the datasets are not yet public (see <https://www.embopress.org/page/journal/14602075/authorguide#datadeposition>).

7) Our journal encourages inclusion of *data citations in the reference list* to directly cite datasets that were re-used and obtained from public databases. Data citations in the article text are distinct from normal bibliographical citations and should directly link to the database records from which the data can be accessed. In the main text, data citations are formatted as follows: "Data ref: Smith et al, 2001" or "Data ref: NCBI Sequence Read Archive PRJNA342805, 2017". In the Reference list, data citations must be labeled with "[DATASET]". A data reference must provide the database name, accession number/identifiers and a resolvable link to the landing page from which the data can be accessed at the end of the reference. Further instructions are available at .

8) At EMBO Press we ask authors to provide source data for the main and EV figures. Our source data coordinator will contact you to discuss which figure panels we would need source data for and will also provide you with helpful tips on how to upload and organize the files.

Numerical data can be provided as individual .xls or .csv files (including a tab describing the data). For 'blots' or microscopy, uncropped images should be submitted (using a zip archive or a single pdf per main figure if multiple images need to be supplied for one panel). Additional information on source data and instruction on how to label the files are available at .

9) We replaced Supplementary Information with Expanded View (EV) Figures and Tables that are collapsible/expandable online (see examples in <https://www.embopress.org/doi/10.15252/embo.201695874>). A maximum of 5 EV Figures can be typeset. EV Figures should be cited as 'Figure EV1, Figure EV2' etc. in the text and their respective legends should be included in the main text after the legends of regular figures.

11) For data quantification: please specify the name of the statistical test used to generate error bars and P values, the number (n) of independent experiments (specify technical or biological replicates) underlying each data point and the test used to calculate p-values in each figure legend. The figure legends should contain a basic description of n, P and the test applied. Graphs must include a description of the bars and the error bars (s.d., s.e.m.).

Please remember: Digital image enhancement is acceptable practice, as long as it accurately represents the original data and conforms to community standards. If a figure has been subjected to significant electronic manipulation, this must be noted in the figure legend or in the 'Materials and Methods' section. The editors reserve the right to request original versions of figures and

the original images that were used to assemble the figure.

We realize that it is difficult to revise to a specific deadline. In the interest of protecting the conceptual advance provided by the work, we recommend a revision within 3 months (24th Jan 2024). Please discuss the revision progress ahead of this time with the editor if you require more time to complete the revisions. Use the link below to submit your revision:

Referee #1:

In this manuscript, the authors investigated the potential role of megakaryocytes (Mks) on hematopoietic stem and progenitor cell development in the embryonic AGM region. They first applied platelet factor 4 (PF4)-Cre;Rosa-tdTomato+ mouse model to show that Mks were enriched in tdTomato+ cells. The authors then conducted a series of experiments to investigate embryonic hematopoiesis after MKs deletion by using PF4-Cre;Rosa-DTA(DTA) embryos, with endpoints of phenotypic HEC, pre-HSC, and HSC quantification, as well as repopulating HSCs. The authors also applied the ex-vivo co-culture system to investigate the role of Mks on endothelial to hematopoietic transition process. scRNA-seq was then performed to characterize Mks, where authors identified Mk progenitors and mature Mks based on the expression of cell surface marker CD226. Co-culturing with HECs with CD226+ Mks, rather than CD226- Mks, promoted the endothelial to hematopoietic transition, with endpoints of total CD45+ cells and CFU-C number. Cell Chat analysis predicted the lineage-receptor interactions between Mks and HECs. Finally, the authors chose TNFSF14 for functional validation experiment, showing that adding TNFSF14 promoted % CD45+ cells, increasing CFU-C number. Overall, the authors identified Mks as potential positive niche cells for embryonic hematopoiesis, in agreement with previous findings of Mks on HSCs in adult BM. However, there are some concerns about this study.

1. The main concern is that no significant difference was found between control and DTA groups in all in vivo transplantation assays, including Fig 3H and Fig 4C&D. In addition, there are only 5 control mice and 3 mice in DTA group in Fig 3H, making it difficult to make a solid conclusion. The authors observed the declined phenotypic HECs and pre-HSCs after Mk deletion. However, cell surface markers cannot completely purify functional HSCs and pre-HSCs. Therefore, as transplantation is the gold-standard assay to quantify functional HSCs, whether Mk deletion affects HSC development is still unclear. Instead of doing more experiments to investigate the role of Mks on functional HSC development, the authors could weaken their statement like 'Mk deletion influences EHT, as evidenced by declined cd45+ cells and CFU number.'
2. Another concern comes from Fig 7. The authors showed that the recombinant TNFSF14 promoted cd45+ cells and CFU numbers in AGM co-culture experiment. However, transplantation experiment is missing, so it is undetermined on HSC development. Second, inflammatory signals are well known for their high compensation. The authors should extend a bit in terms of the downstream signals/targets of TNFSF14 and TNF. Given TNF has already been reported to be a niche signal, this will also help to distinguish TNFSF14 and TNF and highlight the novelty of TNFSF14.
3. In Fig F, the authors reported the additive effects of TNFSF14 and TNF on generating cd45+ cells. However, it is far from being able to conclude 'the synergy between MK and Mac during hematopoiesis in the embryos.' First, the effect of TNFSF14 and TNF is additive, not synergistic. Second, the endpoints are only CD45+ cell numbers, transplantation, and CFU assays are missing. Three, the combined effect of TNFSF14 and TNF is not necessarily related to 'the synergy between MK and Mac', unless more co-culture experiments are performed.

Referee #2:

The manuscript by Lan et al focus on the role of megakaryocytes (Mk) in the onset of HSC embryonic development. They first characterize the presence of Mk by surface markers and show the specific expression of Pf4 to be used as a reporter and a cell deleter in the embryo. Next, they will use genetic DTA expression in these cells to assess the function of Mk in the early hematopoietic development. They find that hemogenic, preHSC and HSCs are affected after deletion of Mk and functional HSCs are also underrepresented. They characterize the cells by single cell RNAseq, identify CD226 as a marker for the functional subpopulation of Mk and show that Tnfsf14 can rescue the Mk depletion effect on hematopoietic production. While Mk are important in the maintenance of adult HSCs, the role of Mk during embryonic hematopoietic development is unknown. Thus, it is a novel and important observation, however some of the experiments are not convincing or at least they should address some issues to rule out some possible misinterpretation of the data.

Major issues:

- 1-There have been previous reports showing that PF4 is also expressed in fetal liver and adult HSCs (PMID: 23300543). This possibility or the fact that it could apply to early HSC development seriously affects most of the conclusions of this work and has to be thoroughly investigated and ruled out. Although authors check the presence of the reporter in AGM (Fig S1D) and linsca+CD201+E 12.5 FL cells, they should further demonstrate that it is not expressed in any of the pre-HSC and HSC

- populations. If this is done by flow cytometry, populations should be sufficiently represented. In addition, the reporter may not cover the same cells as the PF4-cre, thus cre expression in the PF4-Cre;Rosa-DTA HSC or preHSC populations in these embryos should be tested at different times of development from E10.5 HSC-like to E14.5 HSC and HPC development.
- 2- Related to the previous point, the authors do not see any effect in the PF4-Cre;Rosa-DTA E10.5 embryos. Is that also due to expression of PF4 in HSPCs? They should explain why EHT at E10.5 is not affected but it is at E11.5.
 - 3- What is the effect of PF4+ cell deletion in the adult? Can the authors explain if this is not possible due to PF4 expression on adult HSCs?
 - 4- All flow cytometry gates should be shown for all the analysis. Eg. in figure 1A, the Hoechst dots were previously gated for negative cells, but not sure why. Moreover, negative controls for PF4, but specially for CD42d should be included in supplementary.
- Other issues:
- 1- In many figures, the percentage of cells is shown in graphs, however this is a relative number and for examples in case of AGM, the percentage of Mk in total cells depends on the dissection area. They should show number per embryo equivalent shown in other figures.
 - 2- In the yolk sac determinations, are the vitellin/umbilical are included? Please specify.
 - 3- In text corresponding to figure 1, the authors mention the analysis of CD31, but they do not show anywhere. Are Mk CD31? Please show.
 - 4- In figure 1I-1J, most Mk cells are circulating. They should distinguish between circulating and endothelial Mk. In addition, can they test whether Cd226 is different between both populations? An Immunostaining of CD226 would help.
 - 5- Transplantation assays: the engraftment in Fig 3H is very low. Although the reviewer recognizes the challenge of AGM transplants, if this is E11.5 could be improved. The authors use 200000 cells as support, but they may be outcompeting the few HSCs. It is hard to see a clear effect when transplantation is so low. In addition, the transplantation experiments are only meaningful if the expression of PF4 is totally rule out from the HSC population.
 - 6- The authors refer to HEC as CD31+CD44+, to my knowledge these are arterial cells enriched in HECs.
 - 7- English language should be revised
 - 8- Discussion is quite repetitive of results, it should be shortened and focused.

Referee #3:

In this manuscript by Lan and colleagues, the authors use mouse genetic tools to examine a novel role for early megakaryocytes in regulating HSC emergence from hemogenic endothelium. The Pf4-Cre model was used to both label megakaryocytes in the developing embryos (YS, AGM, and FL) as well as delete megakaryocytes by crossing to stop-lox-DTA model. The authors nicely demonstrate that Pf4-labeled MKs are present near hemogenic regions during hematopoietic development, and that deletion of Mks results in a decrease in the frequency of both Mks as well as hematopoietic progenitors in the AGM, particularly at slightly later stages (E11.5). Leveraging scseq analysis, the authors identify two clusters of Mks, CD226+ and CD226-, and determine that the CD226+ Mks that express gene programs related to more mature Mk expression programs are the primary regulators of HSC emergence. Furthermore, they propose that CD226+ Mks regulate HSC emergence via Tnfsf14 secretion. Overall, there are many interesting observations in this manuscript. There are also some aspects that could be clarified further.

In general, there is a lot of variation in the N used for different experiments, and the number of independent experiments is not indicated. For example, in Fig. 2 N is quite high whereas in Fig 3A-E and some of supplemental Figures (S2, S4), N is quite low (N = 4), and it's unclear across experiments (particularly for experiments with low N) if data represent a single experiment/litter. This should be clarified and the use of multiple independent experiments/litters would strengthen the data in cases where N is low. Statistical tests being used should also be indicated for all figures.

In Figure 3, it is unclear why cKIT is not being used to define HSCs and HSPCs. That is standard of the field.

Also in Figure 3H, it is very difficult to draw conclusions from such a low number of mice transplanted (2/5 vs 1/3, where 5% is the metric for engraftment - this metric is very high). Furthermore, in the secondary transplants, it is unclear which primary recipient(s) were selected for secondary transplant. If only one primary recipient was selected, then, it is possible that the single recipient in the control group with the highest chimerism in primary accounts for differences in secondary (which are also very close, generally). These experiments need to be clarified and/or performed with better controls.

In Figure 4, clusters are defined as CD34 Runx1/CKit. Are they both? Either? It is unclear from the text.

In the text referring to Fig. 6, when discussing CD226 expression, data are discussed in the results section that are not referred to (e.g. "Flow cytometric analysis displayed that 29.8{plus minus}2.1% of Mks were positive and more than half of Mks were

negative for CD226 in the E11.5 AGM region, which is different from E10.5 Mks and similar to E11.5 fetal liver." It would be helpful if data were referenced as they were mentioned in the text, as well as labeled above data/FACS panels for YS/AGM/FL (Fig 6E, for example, and throughout Fig. 6).

I think Fig S6S is an important piece of data (given that overall Mks don't decrease at E10.5) and the authors may want to consider including it in the main figure!

Perhaps the weakest link of the paper is data in Figure 7. While the authors convincingly demonstrate that CD226+ MKs express *Tnfsf14*, and their data in Fig 6M-O also strongly suggest that Mks are releasing a secreted factor, the data presented stop short of demonstrating definitively that *Tnfsf14* is the factor released by MKs that regulates HSC emergence. Enhancement of HSC emergence or rescue of the DTA phenotype by the addition of *Tnfsf14* is not the same experiment as MK-specific genetic deletion of *Tnfsf14* in vivo. While that experiment is not absolutely required, some of the language in the manuscript and in the discussion should be toned down to reflect that that experiment has not been directly performed.

In the discussion: the AGM explant experiment in Fig. 4 suggested that depletion of Mks regulates B-cell output upon transplant. Early B-cell output may arise from distinct progenitors during development. Perhaps the authors want to speculate on that.

Dear Reviewer and Editor,

Thanks a lot for your kind consideration and critical comments. In this version, we have addressed carefully all the points one by one. Please check these details in the following pages.

Best regards,

Zhuan

Referee #1:

In this manuscript, the authors investigated the potential role of megakaryocytes (Mks) on hematopoietic stem and progenitor cell development in the embryonic AGM region. They first applied platelet factor 4 (PF4)-Cre;Rosa-tdTomato+ mouse model to show that Mks were enriched in tdTomato+ cells. The authors then conducted a series of experiments to investigate embryonic hematopoiesis after Mks deletion by using PF4-Cre;Rosa-DTA (DTA) embryos, with endpoints of phenotypic HEC, pre-HSC, and HSC quantification, as well as repopulating HSCs. The authors also applied the ex-vivo co-culture system to investigate the role of Mks on endothelial to hematopoietic transition process. scRNA-seq was then performed to characterize Mks, where authors identified Mk progenitors and mature Mks based on the expression of cell surface marker CD226. Co-culturing with HECs with CD226+ Mks, rather than CD226- Mks, promoted the endothelial to hematopoietic transition, with endpoints of total CD45+ cells and CFU-C number. Cell Chat analysis predicted the lineage-receptor interactions between Mks and HECs. Finally, the authors chose TNFSF14 for functional validation experiment, showing that adding TNFSF14 promoted % CD45+ cells, increasing CFU-C number. Overall, the authors identified Mks as potential positive niche cells for embryonic hematopoiesis, in agreement with previous findings of Mks on HSCs in adult BM. However, there are some concerns about this study.

1. The main concern is that no significant difference was found between control and DTA groups in all in vivo transplantation assays, including Fig 3H and Fig 4C&D. In addition, there are only 5 control mice and 3 mice in DTA group in Fig 3H, making it difficult to make a solid conclusion. The authors observed the declined phenotypic HECs and pre-HSCs after Mk deletion. However, cell surface markers cannot completely purify functional HSCs and pre-HSCs. Therefore, as transplantation is the gold-standard assay to quantify functional HSCs, whether Mk deletion affects HSC development is still unclear. Instead of doing more experiments to investigate the role of Mks on functional HSC development, the authors could weaken their statement like 'Mk deletion influences EHT, as evidenced by declined cd45+ cells and CFU number.'

Re: Thanks a lot for your comments and kind consideration. Approximately one adult-repopulated HSCs were observed in the AGM region (Medvinsky et al., *Development*, 2011),

and in direct transplantation experiments, normally around 60% of recipients were engrafted even if two embryo equivalent AGM cells were transplanted. Meanwhile, we have added more recipients for testing HSC and the maturation of pre-HSCs into HSCs. In direct transplantation, 3/6 recipients were engrafted (>5%) after 4 weeks transplantation in the control group, but only one out of 5 recipients in the DTA group. After 20 weeks transplantation, no recipients (0/5) were engrafted in the DTA group, but 5/6 recipients in the control group with the average chimerism $19.04 \pm 12.60\%$ (the following Figure A), suggesting the possibility of reduced HSC activity. Please check all the details in Figure 3H and the modified text in the revised manuscript.

Explant cultures are used to test the HSC precursors (including pre-HSCs). 7/8 recipients had positive engraftment in the control group, however, 5/9 recipients were engrafted in the DTA group at 4 weeks post-transplantation. After 16 weeks transplantation, the chimerism was decreased in the DTA group compared to control group (the following Figure B). These data indicate the reduction of pre-HSC to HSC maturation in the DTA group. Please check all the details in the Figure 4C-4D and the modified text in the revised manuscript.

2. Another concern comes from Fig 7. The authors showed that the recombinant TNFSF14 promoted cd45⁺ cells and CFU numbers in AGM co-culture experiment. However, transplantation experiment is missing, so it is undetermined on HSC development. Second, inflammatory signals are well known for their high compensation. The authors should extend a bit in terms of the downstream signals/targets of TNFSF14 and TNF. Given TNF has already been

reported to be a niche signal, this will also help to distinguish TNFSF14 and TNF and highlight the novelty of TNFSF14.

Re: Thanks a lot for your comments. For the first point, We have tried to perform the transplantation by adding the Tnfsf14 into explant cultures. At 4 weeks transplantation, 2 out of 4 recipients were engrafted with chimerism (22.9% and 35.3%) in the DTA group by Tnfsf14 treatment, similar to control group (3/4 engrafted recipients with chimerism 23.7%, 10%, 7%) and one recipient with 16.4% chimerism in the DTA group (the following Figure A).

Furthermore, three out of four recipients were engrafted with high chimerism in the control group, and in the DTA+Tnfsf14 group, one out of three recipients (unfortunately, one recipient died before 16 weeks transplantation) was engrafted with high chimerism (87.9%) after 16 weeks transplantation, however, the chimerism of one positive recipient (1/4) was 7.5% (the following Figure B). Although the rescued function was not obvious, the possible trend existed with the Tnfsf14 treatment in the DTA group compared to the control group, likely due to the limited recipients.

For the second point, yes, inflammatory signals play roles in the biological process by their high compensation. We have shown that $Tn\alpha$ from macrophages is involved in hematopoiesis (Li et al., blood, 2019 and Mariani et al., Immunity, 2019). $Tn\alpha$ by binding $Tnfrsf1\alpha/\beta$ activates the downstream of target genes, p50/RelA, which is linked to the activation of canonical NF κ B signaling pathways. However, in this study, Light (TNfsf14) derived from megakaryocytes, the ligand of LTBR in endothelial cells, regulates p52/RelB, and then activates non-canonical NF κ B signaling pathways(Piao et al., Cells, 2021). Both of them possibly influence hematopoietic development through distinct pathways, although the crossed talk existed in some common signaling pathways(Lu et al., Frontier in Immunology, 2014). We have modified the discussion and please see the details in the revised manuscript.

3. In Fig F, the authors reported the additive effects of TNFSF14 and TNF on generating cd45+ cells. However, it is far from being able to conclude 'the synergy between MK and

Mac during hematopoiesis in the embryos.’ First, the effect of TNFSF14 and TNF is additive, not synergistic. Second, the endpoints are only CD45⁺ cell numbers, transplantation, and CFU assays are missing. Three, the combined effect of TNFSF14 and TNF is not necessarily related to ‘the synergy between MK and Mac’, unless more co-culture experiments are performed.

Re: Thanks a lot for your comments. Indeed, it is true that the effect of Tnfsf14 and Tnf is additive based on the coculture data by Tnfsf14 and/or Tnftreatment. Meanwhile, we performed cocultures combined Mks with Macs, the number of CD45⁺ cells derived from HECs was enhanced at the existence of Mks or Macs compared to control (HEC group, the following Figure A), similar to the treatment of Tnfsf14 and/or Tnfa.

Methylcellulose culture data showed an increase in total CFU-C number from cocultures in the Mks and/or Macs group (the following Figure B) compared to HEC group. However, the existence of Mac and Mk in Triple group failed to enhance the production of CFU-C as well as CD45⁺ cells compared to the group at the presence of each of them. These data suggest the promotion of Mks or Macs in EHT process. Since we didn’t see the obvious enhancement by combining Macs and Mks, no transplantation was performed after coculture. In addition, we have modified the texts in the revised manuscript. Please see the details in Figure EV5I-5J in the revised manuscript.

Referee #2:

The manuscript by Lan et al focus on the role of megakaryocytes (Mk) in the onset of HSC embryonic development. They first characterize the presence of Mk by surface markers and show the specific expression of Pf4 to be used as a reporter and a cell deleter in the embryo. Next, they will use genetic DTA expression in these cells to assess the function of Mk in the early hematopoietic development. They find that hemogenic, preHSC and HSCs are affected after deletion of Mk and functional HSCs are also underrepresented. They

characterize the cells by single cell RNAseq, identify CD226 as a marker for the functional subpopulation of Mk and show that *Tnfrsf14* can rescue the Mk depletion effect on hematopoietic production.

While Mk are important in the maintenance of adult HSCs, the role of Mk during embryonic hematopoietic development is unknown. Thus, it is a novel and important observation, however some of the experiments are not convincing or at least they should address some issues to rule out some possible misinterpretation of the data.

Major issues:

1-There have been previous reports showing that PF4 is also expressed in fetal liver and adult HSCs (PMID: 23300543). This possibility or the fact that it could apply to early HSC development seriously affects most of the conclusions of this work and has to be thoroughly investigated and ruled out. Although authors check the presence of the reporter in AGM (Fig S1D) and *lin⁻sca⁺CD201⁺E 12.5* FL cells, they should further demonstrate that it is not expressed in any of the pre-HSC and HSC populations. If this is done by flow cytometry, populations should be sufficiently represented. In addition, the reporter may not cover the same cells as the PF4-cre, thus cre expression in the PF4-Cre;Rosa-DTA HSC or preHSC populations in these embryos should be tested at different times of development from E10.5 HSC-like to E14.5 HSC and HPC development.

Re: Thanks a lot for your comments. Firstly, we have checked the expression of tdTomato (tdT+) in the earlier stage of hematopoietic development E10.5 PF4-Cre; Rosa-Tdt AGM region. The percentage and cell number of Tdt⁺ cells in the EC and pre-HSCs are hardly detected (the following Figure A-C), in line with E11.5 AGM cells(Figure 1G-1H).

Secondly, according to your suggestion, we have checked the expression of HSCs in the fetal liver from E12.5 to E14.5 by using the cocktails of HSC (E12.5 HSCs, Lin⁻Sca-1⁺Mac-1^{low}CD201⁺; E13.5-14.5 HSCs, Lin⁻Sca-1⁺c-Kit⁺CD150⁺CD48⁻). TdT⁺ cells were found in the E12.5-14.5 fetal liver (the following Figure A-B). A few tdT⁺ HSCs were found in E12.5-14.5 fetal liver, although the percentage and cell number of tdT⁺ HSCs was very low (the following Figure C-E), lower than that in the previous report (Calaminus et al., *PLOS one*, 2012). In the fetal liver and further late stage, we can't exclude the effects of PF4-Cre labeling HSCs non-specifically.

Thirdly, you are right. PF4-Cre labels distinct cells between AGM and fetal liver. In the AGM region, PF4-Cre is specifically expressed in the MKs according to our data. However, in the fetal liver, PF4-Cre;tdT labels not only Megakaryocytes but also some parts of HSCs from E12.5-14.5 fetal liver.

2-Related to the previous point, the authors do not see any effect in the PF4-Cre;Rosa-DTA E10.5 embryos. Is that also due to expression of PF4 in HSPCs? They should explain why EHT at E10.5 is not affected but it is at E11.5.

Re: Thanks a lot for your critical comments. As we showed in the last comment above, no tdT⁺ signals were found in the E10.5-E11.5 ECs and pre-HSCs of AGM region (in the following Figure

A-C and Figure 1G-1H). All these data are in Figure 1G-1H and Figure EV 1H-1J in the revised manuscript.

From our data, the percentage and cell number of Mks were very low and decreased from E10.5 to E11.5 in the AGM of PF4-Cre;RosaDTA compared to the control group in Figure 2A-2G. The reduction started in the E10.5 AGM region, but not in the E9.5 AGM. And the effects of Mks on the hematopoiesis developed gradually and need some time. Especially, our data showed that the more mature Mks (CD226⁺ Mks) played roles in the EHT. Mks require a mature pattern and then affect the EHT process depending on the time course. That is the reason we didn't observe the change in the E10.5 or even earlier stage of AGM region. More details were modified in the revised manuscript.

3- What is the effect of PF4⁺ cell deletion in the adult? Can the authors explain if this is not possible due to PF4 expression on adult HSCs?

Re: Thanks a lot for your critical comments. As we mentioned above (comments 1-2), although some HSCs in the fetal liver (E12.5-14.5) were positive for tdT⁺;PF4-Cre, which is lower than that in the previous publication (Calaminus et al., *PLOS one*, 2012), but we can't exclude the effects by PF4-non specific labeling HSCs. Calaminus et al. reported that a higher percentage of HSCs from bone marrow expressed tdT⁺, along with our data, we didn't continue to check the expression in bone marrow. The effects of PF4 deletion will be not specific to Mks in bone marrow.

Although PF4-Cre is not specific to HSCs in the bone marrow, it is specific to Mks in the embryos, which is not expressed in the pre-HSC/HSCs in the AGM region. Taken together, PF4-Cre is useful for checking the Mks in the earlier stage of embryos, not suitable for adults.

4- All flow cytometry gates should be shown for all the analysis. Eg. in figure 1A, the Hoechst dots were previously gated for negative cells, but not sure why. Moreover, negative controls for PF4, but specially for CD42d should be included in supplementary.

Re: Thanks a lot for your kind comments. We gated live cells by Hoechst negative and showed the expression of Hoechst and other antibodies. To be more clear, we have labeled the live cell fractions by Hoechst⁺ in Figure 1A.

According to your suggestion, flow analysis data was added as the negative control and PF4-Cre;tdT AGM regions(in the following Figures A and B, respectively). These data are in Figure EV1D, please check all the details in the revised manuscript.

Other issues:

1- In many figures, the percentage of cells is shown in graphs, however this is a relative number and for examples in case of AGM, the percentage of Mk in total cells depends on the dissection area. They should show number per embryo equivalent shown in other figures.

Re: Thanks a lot for your comments. The total cell number of PF4-Cre;Rosa-tdT AGM and yolk sac was analyzed (in the following Figure A-B), which was added to Figure EV1A. Additionally, we have shown the cell number of AGM, YS, FL, which is comparable between control and DTA group (AGM in Figure 2A, YS in Figure EV2A and E10.5-E12.5 FL in Figure EV2B).

2- In the yolk sac determinations, are the vitellin/umbilical are included? Please specify.

Re: Thanks a lot for your comments. We have excluded the vitellin/umbilical vessels in the yolk sac dissection. We have modified the methods in the revised manuscript.

3- In text corresponding to figure1, the authors mention the analysis of CD31, but they do not show anywhere. Are Mk CD31? Please show.

Re: Thanks a lot for your comments. We used CD31 for identifying the endothelial cells, not for Mks. We have modified the texts clearly in the revised manuscript.

4- In figure 1I-1J, most Mk cells are circulating. They should distinguish between circulating and endothelial Mk. In addition, can they test whether Cd226 is different between both populations? An Immunostaining of CD226 would help.

Re: Thanks a lot for your comments. Do you mean that endothelial Mks is the Mk that interacted with endothelial cells, which is distinguished from circulating Mks? We have distinguished each of them from our immunostaining data. From immunostaining data, a very low percentage of Mks (3/82) were contacted with endothelial cells, but most of Mks (79/82) was non-contacted with endothelial cells, similar to platelets (contacted vs non-contacted, 39/255 vs 216/255) (in the following Figure A). These data indirectly support that Mk affects the EHT process through secreted factors. Please check all the details in the Figure EV1M in the revised manuscript.

Additionally, We have checked the subfraction of CD226⁺ Mks by immunostaining for CD226 in the PF4-Cre;Rosa-tdT AGM region. Immunostaining data showed the percentage of CD226⁺

RFP⁺ Mks localized in the DA, which is higher than the percentage of CD226⁻ RFP⁺ Mks. Consistent with above data, rare CD226⁺RFP⁺ Mks close to endothelial cells (the following Figure B-D). So most Mks are circulating cells and CD226 fails to distinguish them due to few Mks interacting with endothelial cells. Please see all the details in Figure EV 4L-4N in the revised manuscript.

5- Transplantation assays: the engraftment in Fig 3H is very low. Although the reviewer recognizes the challenge of AGM transplants, if this is E11.5 could be improved. The authors use 200000 cells as support, but they may be outcompeting the few HSCs. It is hard to see a clear effect when transplantation is so low. In addition, the transplantation experiments are only meaningful if the expression of PF4 is totally rule out from the HSC population.

Re: Thanks a lot for your comments. This is the similar to the comment 1 from reviewer #1. Yes, we have added 200 K bone marrow cells as carrier cells for transplantation, which including about 10 HSCs, much higher than that in the AGM region. That is the reason the chimerism is low. More transplanted recipients were added, you can see the following Figure A-B. In direct transplantation, 3/6 recipients were engrafted (>5%) after 4 weeks transplantation in the control

group, but only one out of 5 recipients in the DTA group. After 20 weeks transplantation, no recipients (0/5) were engrafted in the DTA group, but 5/6 recipients in the control group with the average chimerism $19.04 \pm 12.60\%$ (the following Figure A), suggesting the possibility of reduced HSC activity. Please check all the details in Figure 3H and the modified text in the revised manuscript.

Explant cultures are used to test the HSC precursors (including pre-HSCs). 7/8 recipients had positive engraftment in the control group, however, 5/9 recipients were engrafted in the DTA group at 4 weeks post-transplantation. After 16 weeks transplantation, the chimerism was decreased in the DTA group compared to control group (the following Figure B). These data indicate the reduction of pre-HSC to HSC maturation in the DTA group. Please check all the details in the Figure 4C-4D and the modified text in the revised manuscript.

According to your comment #2 above, We have checked the expression of PF4 in PF4-Cre;RosatdT and have shown that PF4 is not expressed in the pre-HSC and HSCs in the E10.5-E11.5 AGM region (the following Figure A-C and Figure 1G-1H). Along with transplantation data from AGM explant cultures, it makes sense that Mks influence HSC/pre-HSC activity in the AGM region.

6- The authors refer to HEC as CD31+CD44+, to my knowledge these are arterial cells enriched in HECs.

Re: Thanks a lot for your comments. Yes, it is reported that CD44 is an arterial marker in most tissues. However, CD44 is also reported as a hemogenic endothelial cell marker for enriching HEC in the AGM region(Hou et al., *Cell Research* 2020, Oatley et al., *Nature Communications*, 2020). Meanwhile, c-Kit was added to enrich HECs in the E10.5-11.5 AGM region (the following Figure A-D) and E10.5 AGM explant cultures (the following Figure E-F), the changing trend is similar to the previous analysis without c-Kit. Data was updated in Figure 3I, 3L and Figure EV 3E-3F and 3I-3J in the revised manuscript.

7- English language should be revised

Re: Thanks a lot for your comments. We have modified the English language by asking for the help of English native speakers.

8- Discussion is quite repetitive of results, it should be shortened and focused.

Re: Thanks a lot for your comments. We have modified and shortened the discussion.

Referee #3:

In this manuscript by Lan and colleagues, the authors use mouse genetic tools to examine a novel role for early megakaryocytes in regulating HSC emergence from hemogenic endothelium. The Pf4-Cre model was used to both label megakaryocytes in the developing embryos (YS, AGM, and FL) as well as delete megakaryocytes by crossing to stop-lox-DTA model. The authors nicely demonstrate that Pf4-labeled MKs are present near hemogenic regions during hematopoietic development, and that deletion of Mks results in a decrease in the frequency of both Mks as well as hematopoietic progenitors in the AGM, particularly at slightly later stages (E11.5). Leveraging scseq analysis, the authors identify two clusters of Mks, CD226+ and CD226-, and determine that the CD226+ Mks that express gene programs related to more mature Mk expression programs are the primary regulators of HSC emergence. Furthermore, they propose that CD226+ Mks regulate HSC emergence via Tnfsf14 secretion. Overall, there are many interesting observations in this manuscript. There are also some aspects that could be clarified further.

In general, there is a lot of variation in the N used for different experiments, and the number of independent experiments is not indicated. For example, in Fig. 2 N is quite high whereas in Fig 3A-E and some of supplemental Figures (S2, S4), N is quite low (N = 4), and it's unclear across experiments (particularly for experiments with low N) if data represent a single experiment/litter. This should be clarified and the use of multiple independent

experiments/litters would strengthen the data in cases where N is low. Statistical tests being used should also be indicated for all figures.

Re: Thanks a lot for your comments. In Fig2, we have shown the embryos from more than 8 times experiments. Because the percentage is very low in the pre-HSC I and II from AGM region, we repeated more times than that in fetal liver. Meanwhile, we also repeated these experiments for fetal liver analysis (the following Figure A-F). Please see the details in Figure 3A-3F. Additionally, we added all the details in methods. The experimental times, and statistics were shown in the figure legends. Please see all the details in the revised manuscript.

In Figure 3, it is unclear why cKIT is not being used to define HSCs and HSPCs. That is standard of the field.

Re: Thanks a lot for your comments. This comment is similar to the comment #6 from Reviewer 2. In the AGM region, we have analyzed the pre-HSC by using c-Kit and we also added c-Kit in the HEC analysis in the revised manuscript. C-Kit was added to enrich HECs in the E10.5-11.5 AGM region (the following Figure A-D) and E10.5 AGM explant cultures (the following Figure E-F), the change trend is similar to the previous analysis without c-Kit. Data was updated in Figures 3I, 3L and Figure EV 3E-3F and 3I-3J in the revised manuscript.

For the E12.5 fetal liver, we used CD201+LSM and HSC by Lin, Sca1, and Mac1 instead of LSK(Lin⁻c-Kit⁺Sca1⁺) according to the published paper(Zhou et al., *Nature*, 2016).

Also in Figure 3H, it is very difficult to draw conclusions from such a low number of mice transplanted (2/5 vs 1/3, where 5% is the metric for engraftment - this metric is very high). Furthermore, in the secondary transplants, it is unclear which primary recipient(s) were selected for secondary transplant. If only one primary recipient was selected, then, it is possible that the single recipient in the control group with the highest chimerism in primary accounts for differences in secondary (which are also very close, generally). These experiments need to be clarified and/or performed with better controls.

Re: Thanks a lot for your comments. This is a similar comment to that from Reviewers 1 and 2. Approximately one adult-repopulated HSCs were observed in the AGM region (Medvinsky et al., *Development*, 2011), and in direct transplantation experiments, normally around 60% of recipients were engrafted even if two embryo equivalent AGM cells were transplanted. Meanwhile, we have added more recipients for testing HSC and the maturation of pre-HSCs into HSCs. In direct transplantation, 3/6 recipients were engrafted (>5%) after 4 weeks transplantation in the control group, but only one out of 5 recipients in the DTA group. After 20 weeks transplantation, no recipients (0/5) were engrafted in the DTA group, but 5/6 recipients in the control group with the average chimerism $19.04 \pm 12.60\%$ (the following Figure A), suggesting the possibility of reduced HSC activity.

Because of no engrafted recipients in the DTA group, we didn't perform the secondary transplantation. Please check all the details in Figure 3H and the modified text in the revised manuscript.

In Figure 4, clusters are defined as CD34 Runx1/CKit. Are they both? Either? It is unclear from the text.

Re: Thanks a lot for your comments. In Figure 4B, clusters were calculated for the number of both CD34⁺Runx1⁺ and CD34⁺c-Kit⁺ clusters. In supplementary Figure 4A, the cell number of CD34⁺Runx1⁺ or CD34⁺c-Kit⁺ cluster was displayed, respectively. The text was modified clearly, please see the modified manuscript.

In the text referring to Fig. 6, when discussing CD226 expression, data are discussed in the results section that are not referred to (e.g. "Flow cytometric analysis displayed that 29.8{plus minus}2.1% of Mks were positive and more than half of Mks were negative for CD226 in the E11.5 AGM region, which is different from E10.5 Mks and similar to E11.5 fetal liver." It would be helpful if data were referenced as they were mentioned in the text, as well as labeled above data/FACS panels for YS/AGM/FL (Fig 6E, for example, and throughout Fig. 6).

Re: Thanks a lot for your comments. We have added Figure 6E-6F and Figure EV40-4Q in the revised manuscript.

I think Fig S6S is an important piece of data (given that overall Mks don't decrease at E10.5) and the authors may want to consider including it in the main figure!

Re: Thanks a lot for your comments. We have moved this figure into Figure 6L in the revised manuscript.

Perhaps the weakest link of the paper is data in Figure 7. While the authors convincingly

demonstrate that CD226+ MKs express Tnfsf14, and their data in Fig 6M-0 also strongly suggest that Mks are releasing a secreted factor, the data presented stop short of demonstrating definitively that Tnfsf14 is the factor released by MKs that regulates HSC emergence. Enhancement of HSC emergence or rescue of the DTA phenotype by the addition of Tnfsf14 is not the same experiment as MK-specific genetic deletion of Tnfsf14 in vivo. While that experiment is not absolutely required, some of the language in the manuscript and in the discussion should be toned down to reflect that that experiment has not been directly performed.

Re: Thanks a lot for your comments. This is similar to the comment 2 from reviewer 1. Yes, we only focus one of secreted factors(Tnfsf14) to test the rescue function in vitro. It's better to establish MK specifically deleted Tnfsf14 mouse model for checking the roles of Tnfsf14 derived from Mks. It is a pity no mouse model is available until now and This will be in plan in future. We have added Tnfsf14 into explant culture for checking the pre-HSC maturation. At 4 weeks transplantation, 2 out of 4 recipients were engrafted with chimerism (22.9% and 35.3%) in the DTA group by Tnfsf14 treatment, similar to control group (3/4 engrafted recipients with chimerism 23.7%, 10%, 7%) and one recipient with 16.4% chimerism in the DTA group (the following Figure A).

Furthermore, three out of four recipients were engrafted with high chimerism in the control group, and in the DTA+Tnfsf14 group, one out of three recipients (unfortunately, one recipient died before 16 weeks transplantation) was engrafted with high chimerism (87.9%) after 16 weeks transplantation, however, the chimerism of one positive recipient (1/4) was 7.5% (the following Figure B). Although the rescued function was not obvious, the possible trend existed with the Tnfsf14 treatment in the DTA group compared to the control group, likely due to the limited recipients. Meanwhile, we have modified the language and shortened the discussion according to your suggestion.

In the discussion: the AGM explant experiment in Fig. 4 suggested that depletion of Mks regulates B-cell output upon transplant. Early B-cell output may arise from distinct progenitors during development. Perhaps the authors want to speculate on that.

Re: Thanks a lot for your comments. You are correct that some parts of B cells derive from other B cell progenitors in the yolk sac during development (Yoshimoto et al., *PNAS*, 2011). We have added more recipients and modified the lineage output in the Figure 4E. We have corrected the text and please see all the details in the revised manuscript.

Dear Dr Zhuan Li,

Thank you for submitting your revised manuscript (EMBOJ-2023-115554R) to The EMBO Journal. Your amended study was sent back to the three referees for their scientific re-evaluation, and we have received detailed comments from all of them, which I enclose below.

As you will see, the experts state that the work has been substantially improved by the revisions and they are now in favour of publication, pending minor revision.

Thus, we are pleased to inform you that your manuscript has been accepted in principle for publication in The EMBO Journal.

Please consider the remaining minor comment of referees #2 and #3 regarding data annotation carefully and amend the manuscript figures and text accordingly.

Also, we now need you to take care of a number of issues related to formatting and data presentation as detailed below, which should be addressed at re-submission.

Please contact me at any time if you have additional questions related to below points.

As you might have seen on our web page, every paper at the EMBO Journal now includes a 'Synopsis', displayed on the html and freely accessible to all readers. The synopsis includes a 'model' figure as well as 2-5 one-short-sentence bullet points that summarize the article. I would appreciate if you could provide this figure and the bullet points.

Thank you for giving us the chance to consider your manuscript for The EMBO Journal. I look forward to your final revision.

Again, please contact me at any time if you need any help or have further questions.

Best regards,

Daniel Klimmeck

>> Author Contributions: Please remove the author contributions information from the manuscript text. Note that CRediT has replaced the traditional author contributions section as of now because it offers a systematic machine-readable author contributions format that allows for more effective research assessment. and use the free text boxes beneath each contributing author's name to add specific details on the author's contribution.

More information is available in our guide to authors.
<https://www.embopress.org/page/journal/14602075/authorguide>

>> Adjust the title of the Competing Interests' section to 'Disclosure and Competing Interests Statement'.

>> Data Availability Section: Remove duplicate Data Availability Section on p. 30-31.

>> Funding: please merge with Acknowledgments, the complete funding information still needs to be entered into our online manuscript system.

>> References: the References section should be placed after the Discussion.

>> Appendix / EV figures: Supplemental figures can either be uploaded as Figures EV1-5: individual, high-resolution figure files, one page per figure, and their legends added to the manuscript text, after the main figure legends, with Table EV1 also uploaded as an individual file; or remain compiled in one PDF file with their as "Appendix Figure S1" - S5. Table EV1 should then be renamed "Appendix Table S1". The appendix PDF would need a table of contents with page numbers.

>> Cite the Material and Methods referenced from Hou et al (2022; PMID: 35079138) in the Material and Methods section.

>> Consider additional changes and comments from our production team as indicated below:

- Figure legends:

1. Please define the annotated p values ***/**/* in the legend of figure 7b as appropriate.

>>> pls check

2. Please indicate the statistical test used for data analysis in the legends of figures 1b-c, e-f, j; 2a-i; 3a-g, i-n; 4b, d, g-h; 5c-d; 6d, f-g, i-m, o-p; 7a-e, g; Ev 1a, c, f-g, l-m, p-q; EV 2a-f, h; EV 3b, d, g-j; EV 4m, EV p-f1; EV 5c, f-j.

>>> pls check

3. Please note that in figures EV 1c, f-g; EV 4D1-E1; there is a mismatch between the annotated p values in the figure legend and the annotated p values in the figure file that should be corrected.

>>> pls check

4. Please note that the box plots need to be defined in terms of minima, maxima, centre, bounds of box and whiskers, and percentile in the legends of figures 6d; 7b; EV 5g.

>>> pls check

5. Please note that information related to n is missing in the legends of figures 6d; 7b; EV 5g.

>>> pls check

6. Although 'n' is provided, please describe the nature of entity for 'n' in the legends of figures 1b-c, e-f, j; 2a-i; 3a-g, i-n; 4b, e, g-h; 6f-g, i-m, o-p; 7c-e, g; EV 1a, c, f-g, j, l-m, p-q; EV 2a-f, h; EV 3a-h, k-l; EV 4m-f1; EV 5c-d, h-j.

>>> pls check

7. Please note that the error bars are not defined in the legends of figures 1b-c, e-f, j; 2a-i; 3a-g, i-n; 4b, e, g-h; 6f-g, i-m, o-p; 7c-e, g; EV 1a, c, 1f-g, l-m, p-q; EV 2a-f, h; EV 3a-l; EV 4m-f1; Ev 5c-d, h-j.

>>> pls check

Please use the link below to submit your revision:

Referee #1:

In this revised MS, the authors have made substantial efforts in addressing our concerns.

Referee #2:

I acknowledge the work that the authors have performed for the revisions of the manuscript and the transplantation data is slightly improved, however the related text should be revised in page 6:

Whereas only 3 out of 6 mice receiving control AGM cells were engrafted and 1 out of 5 recipients were engrafted in the DTA group at 4 weeks post-transplantation.

If the authors have the data of engraftment of Bone Marrow, I would recommend them to include it. BM can show better engraftment.

I also agree that possible expression of PF4 in the HSCs seems not too important.

There is one thing that worries me with the revisions, this is the inclusion of ckit marker in the HEC population. The idea of using

ckit is just to restrict hematopoietic cells to intraortic cluster cells which all express ckit, but, to my knowledge, HECs do not express ckit yet. In any case, it is hard to understand which populations have they measured in FIG3 I-N since the gating strategy has not been included.

Their definition for each subpopulation is:

HEC (CD41-CD45-CD31+CD44+c-Kit+),

pre-HSC I (CD41^{low}CD45-CD31+CD201+c-Kit+),

and pre-HSC II (CD45+CD31+CD201+c-Kit+)

If they use ckit+ for their definition of HECs, they are referring to a similar pre-HSC I population, likely different than the one that includes CD201, but they are excluding HECs. They need to count CD41-CD45-CD31+CD44+ckit- for HECs!

In any case, they should include one example of the gating for all these populations.

Referee #3:

The authors have made a strong effort to address all of the reviewers' previous concerns and have improved the manuscript. Most notably, the authors have increased the number of mice used in transplantation assays and have also clarified number of mice and independent experiments performed throughout the manuscript.

For the transplant data in Figure 3, it's still not entirely clear why a cutoff of 5% engraftment was chosen to indicate "engrafters" - this is a very high cut-off and seems somewhat random. Can the authors instead report total engraftment for both groups? Are these differences statistically significant (or was a test of significance applied?) I think that although these experiments are technically challenging, the data are best presented as transparently as possible.

Data for AMG explant transplants are not correctly referenced in the text (reference is missing).

While the additional data with explant culture transplants in Figure 7 provide some additional support for the author's claim that *tnfsf14* regulates HSC emergence, the experimental numbers for transplantation experiments are still very low - it is very difficult to make a determination from one engrafted experiments vs zero in the DTA only group at 20 weeks. I do feel that the authors still need to directly indicate the limitation of this experiment and the conclusions of these data in their discussion. The final paragraph in their discussion still suggests that they have shown directly that MKs regulate HSC emergence through the production of *tnfsf14*. I think it would be more judicious to state what was discovered and what was not shown directly.

Dear Daniel and Reviewers,

Thanks a lot for your kind consideration and we are so glad to have the chance to revise the manuscript. We have addressed all the points one by one, which are highlighted in red color. Please check all the details on the following pages and in the revised manuscript. The synopsis including “model figure” and bullet points is shown in a separate file.

Please don't hesitate to contact me if you have any questions.

Best,

Zhuan

>> Author Contributions: Please remove the author contributions information from the manuscript text. Note that CRediT has replaced the traditional author contributions section as of now because it offers a systematic machine-readable author contributions format that allows for more effective research assessment. and use the free text boxes beneath each contributing author's name to add specific details on the author's contribution.

Re: We have done it.

More information is available in our guide to authors.

>> Adjust the title of the Competing Interests' section to 'Disclosure and Competing Interests Statement'.

Re: We have done it.

>> Data Availability Section: Remove duplicate Data Availability Section on p. 30-31.

Re: We have done it.

>> Funding: please merge with Acknowledgments, the complete funding information still needs to be entered into our online manuscript system.

Re: We have done it from the online system.

>> References: the References section should be placed after the Discussion.

Re: We have done it.

>> Appendix / EV figures: Supplemental figures can either be uploaded as Figures EV1-5: individual, high-resolution figure files, one page per figure, and their legends added to the manuscript text, after the main figure legends, with Table EV1 also uploaded as an individual file; or remain compiled in

one PDF file with their as "Appendix Figure S1" - S5. Table EV1 should then be renamed "Appendix Table S1". The appendix PDF would need a table of contents with page numbers.

Re: We have kept one PDF file with their as "Appendix Figure S1 - S5" and renamed all of them according the request.

>> Cite the Material and Methods referenced from Hou et al (2022; PMID: 35079138) in the Material and Methods section.

Re: We have done it.

>> Consider additional changes and comments from our production team as indicated below:

- Figure legends:

1. Please define the annotated p values *****/**/*** in the legend of figure 7b as appropriate.

>>> pls check

Re: We have checked all of them and no changes have been made since we modified them in the last version of the manuscript submitted on 21st, Jan, 2024.

2. Please indicate the statistical test used for data analysis in the legends of figures 1b-c, e-f, j; 2a-i; 3a-g, i-n; 4b, d, g-h; 5c-d; 6d, f-g, i-m, o-p; 7a-e, g; Ev 1a, c, f-g, l-m, p-q; EV 2a-f, h; EV 3b, d, g-j; EV 4m, EV p-f1; EV 5c, f-j.

>>> pls check

Re: We have checked all of them and no changes have been made since we modified them in the last version of the manuscript submitted on 21st, Jan, 2024.

3. Please note that in figures EV 1c, f-g; EV 4D1-E1; there is a mismatch between the annotated p values in the figure legend and the annotated p values in the figure file that should be corrected.

>>> pls check

Re: We have checked all of them and no changes have been made since we modified them in the last version of the manuscript submitted on 21st, Jan, 2024.

4. Please note that the box plots need to be defined in terms of minima, maxima, centre, bounds of box and whiskers, and percentile in the legends of figures 6d; 7b; EV 5g.

>>> pls check

Re: We have checked all of them and no changes have been made since we modified them in the last version of the manuscript submitted on 21st, Jan, 2024.

5. Please note that information related to n is missing in the legends of figures 6d; 7b; EV 5g.

>>> pls check

Re: We have checked all of them and no changes have been made since we modified them in the last version of the manuscript submitted on 21st, Jan, 2024.

6. Although 'n' is provided, please describe the nature of entity for 'n' in the legends of figures 1b-c, e-f, j; 2a-i; 3a-g, i-n; 4b, e, g-h; 6f-g, i-m, o-p; 7c-e, g; EV 1a, c, f-g, j, l-m, p-q; EV 2a-f, h; EV 3a-h, k-l; EV 4m-f1; EV 5c-d, h-j.

>>> pls check

Re: We have checked all of them and no changes have been made since we modified them in the last version of the manuscript submitted on 21st, Jan, 2024.

7. Please note that the error bars are not defined in the legends of figures 1b-c, e-f, j; 2a-i; 3a-g, i-n; 4b, e, g-h; 6f-g, i-m, o-p; 7c-e, g; EV 1a, c, 1f-g, l-m, p-q; EV 2a-f, h; EV 3a-l; EV 4m-f1; Ev 5c-d, h-j.

>>> pls check

Re: We have checked all of them and no changes have been made since we modified them in the last version of the manuscript submitted on 21st, Jan, 2024.

Referee #1:

In this revised MS, the authors have made substantial efforts in addressing our concerns.

Re: Thanks a lot.

Referee #2:

I acknowledge the work that the authors have performed for the revisions of the manuscript and the transplantation data is slightly improved, however the related text should be revised in page 6:

Whereas only 3 out of 6 mice receiving control AGM cells were engrafted and 1 out of 5 recipients were engrafted in the DTA group at 4 weeks post-transplantation.

If the authors have the data of engraftment of Bone Marrow, I would recommend them to include it. BM can show better engraftment.

Re: Thanks a lot for your suggestion. Since no recipients were engrafted in the DTA group, we didn't continue to check the chimerism in the bone marrow. We have added the average chimerism of all the recipients in the revised manuscript, please check all the details.

I also agree that possible expression of PF4 in the HSCs seems not too important.

Re: Thanks a lot. We do agree with you.

There is one thing that worries me with the revisions, this is the inclusion of ckit marker in the HEC population. The idea of using ckit is just to restrict hematopoietic cells to intraortic cluster cells which all express ckit, but, to my knowledge, HECs do not express ckit yet. In any case, it is hard to understand which populations have they measured in FIG3 I-N since the gating strategy has not been included.

Their definition for each subpopulation is:

HEC (CD41-CD45-CD31+CD44+c-Kit+),

pre-HSC I (CD41^{low}CD45-CD31+CD201+c-Kit+),

and pre-HSC II (CD45+CD31+CD201+c-Kit+)

If they use ckit+ for their definition of HECs, they are referring to a similar pre-HSC I population, likely different than the one that includes CD201, but they are excluding HECs. They need to count CD41-CD45-CD31+CD44+ckit- for HECs!

In any case, they should include one example of the gating for all these populations.

Re: Thanks a lot for your critical point. We also used these markers to analyze HEC (CD41-

CD45⁺CD31⁺CD44⁺c-Kit⁺), which can distinguish from pre-HSC I (CD41^{low}CD45⁺CD31⁺CD201⁺c-Kit⁺) based on the different expression of CD41. For CD41⁻CD45⁺CD31⁺CD44⁺ckit⁻ for HECs, the trend is similar to cKit⁺ HEC. As reported by previous publication (Hou et al., Cell Research, 2020), the cocktail of CD201(Procr), c-Kit, CD44, is used for enriching the HEC. We have used the same markers for enrichment of HECs. The presentative flow analysis data(the following figure) have been added in Figure EV3E and all the data related to HECs have been modified (Figure 3I, 3L and Appendix Fig. S3E-EV3G, S3J-3K).

Referee #3:

The authors have made a strong effort to address all of the reviewers' previous concerns and have improved the manuscript. Most notably, the authors have increased the number of mice used in transplantation assays and have also clarified number of mice and independent experiments performed throughout the manuscript.

For the transplant data in Figure 3, it's still not entirely clear why a cutoff of 5% engraftment was chosen to indicate "engrafters" - this is a very high cut-off and seems somewhat random. Can the authors instead report total engraftment for both groups? Are these differences statistically significant (or was a test of significance applied?) I think that although these experiments are technically challenging, the data are best presented as transparently as possible.

Re: Thanks a lot for your critical point. Firstly, the chimerism 5% was chosen according to our previous publications (Mariani et al., Immunity, 2019 and Li et al, Blood, 2019). Secondly, We have shown all the chimerism in the Figure 3H for the control and DTA group because one circle or triangle signal presents one recipient. The average chimerism was added in the

revised manuscript. Thirdly, no significant statistics were found between both groups based on all chimerism data, that is why we didn't add any statistic asterisk.

Data for AMG explant transplants are not correctly referenced in the text (reference is missing).

Re: Thanks a lot for your suggestion. We have modified them.

While the additional data with explant culture transplants in Figure 7 provide some additional support for the author's claim that *tnfsf14* regulates HSC emergence, the experimental numbers for transplantation experiments are still very low - it is very difficult to make a determination from one engrafted experiments vs zero in the DTA only group at 20 weeks. I do feel that the authors still need to directly indicate the limitation of this experiment and the conclusions of these data in their discussion. The final paragraph in their discussion still suggests that they have shown directly that MKs regulate HSC emergence through the production of *tnfsf14*. I think it would be more judicious to state what was discovered and what was not shown directly.

Re: Thanks a lot for your critical point. We have modified the related text. We emphasized HPC, but not HSCs.

Dear Dr Zhuan Li,

Thank you for submitting the revised version of your manuscript. I have now evaluated your amended manuscript and concluded that the remaining minor concerns have been sufficiently addressed.

I am pleased to inform you that your manuscript has been accepted for publication in the EMBO Journal.

On a different note, I would like to alert you that EMBO Press offers a format for a video-synopsis of work published with us, which essentially is a short, author-generated film explaining the core findings in hand drawings, and, as we believe, can be very useful to increase visibility of the work. Please see the following link for representative examples and their integration into the article web page:

<https://www.embopress.org/doi/full/10.15252/emboj.2019103932>

with
Best regards,

Daniel Klimmeck

Daniel Klimmeck, PhD
Senior Editor
The EMBO Journal
EMBO
Postfach 1022-40
Meyerhofstrasse 1
D-69117 Heidelberg
contact@embojournal.org
Submit at: <http://emboj.msubmit.net>
